# Co-variability of the summer NDVIs on the eastern Tibetan Plateau and in the Lake Baikal region: Associated climate factors and atmospheric circulation

Kejun He[1], Ge Liu[1,2]*, Junfang Zhao[1]*, Jingxin Li[1]

**1** State Key Laboratory of Severe Weather, Chinese Academy of Meteorological Sciences, Beijing, China,
**2** Collaborative Innovation Centre on Forecast and Evaluation of Meteorological Disasters, Nanjing University of Information Science and Technology, Nanjing, China

\* liuge@cma.gov.cn (GL); zhaojfcams@163.com (JZ)

**Data Availability Statement:** All relevant data are within the manuscript and its Supporting Information files.

## Abstract

The Tibetan Plateau and Siberia are both crucial regions in which the vegetation dynamics are sensitive to climate change. The variabilities in the normalized difference vegetation index (NDVI) over the two regions have been explored previously, but there have been few studies on the relationship of the NDVI in the two regions. Using the GIMMS-NDVI, GHCN-CAMS and NCEP reanalysis datasets and statistical and physical diagnostic methods, we show that the summer (June, July and August) NDVI over the eastern Tibetan Plateau and Lake Baikal and its adjacent eastern region of Siberia have an in-phase co-variability, especially on an interannual timescale (with a correlation coefficient of 0.69 during the time period 1982–2014). Further analyses show that precipitation and the related cloud cover and solar radiation are responsible for the variability in the NDVI over the eastern Tibetan Plateau, whereas temperature has the more important role in modulating the variability in the NDVI over the Lake Baikal region. A dipole pattern prevails over the Tibetan Plateau–Lake Baikal region and reflects the anomalies in the intensity and location of the South Asian high and the northeast Asian blocking high. This dipole pattern simultaneously modulates precipitation over the eastern Tibetan Plateau and the temperature over the Lake Baikal region and leads to the co-variability of the NDVI between the two regions. A synergistic sea surface temperature index, which reflects sea surface temperature anomalies in the eastern tropical Pacific Ocean, the northwest Pacific Ocean, the northern Indian Ocean and the subtropical north Atlantic Ocean, appears to adjust this Tibetan Plateau–Lake Baikal dipole pattern and is therefore closely related to the co-variability of the NDVI between the eastern Tibetan Plateau and the Lake Baikal region. Our results suggest that vegetation dynamics may not be only a local phenomenon in some areas, but are also likely to remotely link with variations in vegetation over other regions.

**Funding:** The National Key Research and Development Program of China (2018YFC1505706) is replaced with the Second Tibetan Plateau Scientific Expedition and Research (STEP) program (2019QZKK0105) This research is funded by the National Key Research and Development Program of China (2018YFC1505706), the Strategic Priority Research Program of the Chinese Academy of Sciences (XDA20100300), the National Science Foundation of China (91637312 and 41775084), the Science and Technology Development Fund of CAMS (2019KJ022), and the Basic Research Fund of CAMS (2019Z008). The funders had no role in study design, data collection and analysis, decision to publish, or preparation of the manuscript.

**Competing interests:** The authors have declared that no competing interests exist.

# Introduction

The Tibetan Plateau, located in the subtropical region of eastern Eurasia, is the highest plateau in the world. The Tibetan Plateau is extremely sensitive to climate change as a result of its high elevation and unique geographical location [1]. Vegetation on the Tibetan Plateau is highly susceptible and vulnerable to global climate change [2,3]. Analysis of the normalized difference vegetation index (NDVI) and monitoring of the vegetation cover, growth and photosynthetic intensity [4] has shown that the vegetation on the Tibetan Plateau has experienced increasing trends in recent years [5–8]. Temperature and precipitation are considered to be the dominant climate factors affecting variations in the NDVI over the Tibetan Plateau [2,7,9–11]. In recent years, surface air temperature over the Tibetan Plateau showed a significant increasing trend [12,13]. There was also an increasing trend in precipitation over the Tibetan Plateau [14,15]. In general, an appropriate increase in temperature and precipitation may facilitate the growth of vegetation. However, the vegetation types gradually change from forest and shrub over the southeastern Tibetan Plateau to meadow, grass, and desert vegetation over the northwestern Tibetan Plateau [7]. This diversity of vegetation types causes various responses of the NDVI to climate change. As such, in the Tibetan Plateau, there is a clear temporal and spatial heterogeneity in the relationship between the NVDI and climate factors [2,7,16,17]. The climate factors governing the variations in the NDVI over specific regions of the Tibetan Plateau therefore require further investigation.

Siberia is located in the mid- and high latitudes of Eurasia to the north of the Tibetan Plateau and is another region where the variation in vegetation is very sensitive to climate change [18]. Siberia is covered mainly by forests and woodlands [19], which is generally similar to the southeastern Tibetan Plateau though different from the northwestern Tibetan Plateau. Surface air temperature is considered as the major factor influencing vegetation over the Siberia [4,20]. Previous studies revealed that as a result of pronounced warming, the NDVI over the mid- and high latitudes of Eurasia, centered on Siberia, has experienced an increasing trend since the early 1980s [4,21–24], which seems to be consistent with the increasing trend in vegetation on the Tibetan Plateau [5–8].

Given the similarities of vegetation types and climate background over Siberia and the southeastern Tibetan Plateau and the sensitivities of the NDVIs over the two regions to climate change, it is necessary to explore whether the NDVI over the Tibetan Plateau is connected with that over Siberia during summer. The present study found an in-phase co-variability of the summer NDVI between the eastern Tibetan Plateau and Lake Baikal and its adjacent eastern region of Siberia. More interestingly, we detected that this co-variability is primarily presented on an interannual timescale, which implies that this co-variability should not be simply attributed to the decadal trend under a background of global warming. As such, we further explore the possible reason for this co-variability from the perspectives of climate factors and atmospheric circulation.

The characteristics of the NDVI over the Tibetan Plateau and Siberia and their corresponding climate factors have been widely studied, respectively [9,25–29]. To our knowledge, there is no study on the relationship between the NDVI over the two regions. The co-variability of the summer NDVI between the two regions is a new finding. Investigating climate factors and atmospheric circulation related to this co-variability may improve the understanding of the response of regional vegetation to climate change and clarify the reason for the synergistic change of vegetation in different regions. This study is possibly conducive to recognizing the contribution of atmospheric circulation and associated climate factors to the terrestrial ecosystem from a new aspect (i.e., the co-variability of vegetation), and therefore may provide a

scientific basis for the organization and cooperation of ecological and environmental protection among countries and areas.

## Materials and methods

### Materials

The NDVI was used to study the variations in vegetation over the Tibetan Plateau and Lake Baikal region. As an important source of information for the qualitative and quantitative evaluation of vegetation coverage and growth activity [8,30], the NDVI has been widely applied in research on vegetation dynamics across the globe [31–33]. The NDVI used in this study was obtained from the GIMMS-NDVI 3g.v1 dataset based on measurements from the Advanced Very High-Resolution Radiometer (AVHRR) mounted on the National Oceanic and Atmospheric Administration (NOAA) series satellites and provided by NASA's Global Inventor Modeling and Mapping Studies (GIMMS). The spatial resolution of the NVDI dataset is 8 km × 8 km and its time span is from 1982 to 2014. Several studies have documented that this NDVI is one of the best sources of data with which to analyze vegetation dynamics on long-term timescales [34–36]. It eliminates, to a great extent, the effects irrelevant to variations in vegetation and reduces systematic and periodic errors [35,37–40]. This dataset is therefore suitable for studies of the Tibetan Plateau, a region with a large spatial range and diverse climate types [28,41]. To eliminate the effects of soil (over deserts and sparsely vegetated grids), snow, and water bodies, the NDVIs lower than 0.1 should not be considered [7]. There is almost no NVDI lower than 0.1 over the study areas of this paper (see the green boxes in Fig 2), implying that the disturbance of water body/bare ground on the NDVI has been removed in this dataset.

The maximum value composite method [42] was used to calculate the monthly NDVI values during the time period 1982–2014. To match the meteorological datasets (e.g., the 2-m air temperature), the monthly NDVI dataset was interpolated into 0.5˚ × 0.5˚ grids. Previous studies have pointed that the distribution of summer (June–August) precipitation over the Tibetan Plateau closely resembles the annual pattern, that is, summer precipitation makes the largest contribution to annual precipitation on the Tibetan Plateau, accounting for ∼60–70% of annual total and affects the vegetation gradient across the Tibetan Plateau [43,44]. This study primarily investigated the variation in the NDVI during the summer months because this is the most important period for the growth of vegetation on the Tibetan Plateau.

Meteorological datasets were used to explore the climate factors and atmospheric circulation and sea surface temperature anomalies (SSTAs) affecting the NDVI, including the monthly mean precipitation dataset from the Global Precipitation Climatology Centre (GPCC v2018) with a spatial resolution of 0.5˚ × 0.5˚ [45], the monthly mean 2-m air temperature dataset derived from the GHCN_CAMS Reanalysis Products of the National Centers for Environmental Prediction (NCEP) on 0.5˚ × 0.5˚ grids [46], the sea level pressure and standard level geopotential height and *U-* and *V-* wind datasets from the CDC Reanalysis Products of the NCEP at a spatial resolution of 2.5˚ × 2.5˚ [47] and the SST on a 2˚ × 2˚ grid obtained from the NOAA Extended Reconstructed Sea Surface Temperature Version 4 dataset [48]. We also used the NOAA 2.5˚ × 2.5˚ interpolated outgoing longwave radiation (OLR) [49], which reflects the cloud cover, with lower (higher) OLR corresponding to stronger (weaker) convection/more (less) cloud. Also, the OLR is closely related to precipitation anomalies and vegetation coverage [50]. The downward shortwave radiation flux on a resolution of 192×94 Gaussian grid was obtained from the NCEP/NCAR Reanalysis dataset [47], which reflects the solar radiation received by the Earth's surface. All these datasets were extracted for the same time period (1982–2014) as the NDVI dataset.

## Methods

An empirical orthogonal function (EOF) [51] was used to acquire the dominant mode of atmospheric circulation and to explore the relationship between the NDVI and typical atmospheric circulation patterns. The EOF method can be used to decompose the original variables into linear combinations of orthogonal functions and generates irrelative typical modes containing as much information as possible about the original variables. This method has been widely used to analyze the characteristics of climate variable fields since it was introduced into the atmospheric sciences by Obukhov [52]. The EOF method is presented in detail as follows. It can decompose a climate variable field in matrix form:

$$X = \begin{bmatrix} x_{11} & x_{12} & \cdots & x_{1j} & \cdots & x_{1n} \\ x_{21} & x_{22} & \cdots & x_{2j} & \cdots & x_{2n} \\ \vdots & \vdots & & \vdots & & \vdots \\ x_{m1} & x_{m2} & \cdots & x_{mj} & \cdots & x_{mn} \end{bmatrix} \tag{1}$$

in which $m$ represents the spatial point, $n$ represents the time point and $x_{mn}$ represents the $n^{th}$ observed value on the $m^{th}$ station or grid, respectively.

Through an EOF expansion, the above matrix can be decomposed into the spatial function and time function:

$$x_{ij} = \sum_{k=1}^{m} v_{ik} t_{kj} = v_{i1} t_{1j} + v_{i2} t_{2j} + \cdots + v_{im} t_{mj} \tag{2}$$

or

$$X = VT, \tag{3}$$

in which $V = \begin{bmatrix} v_{11} & v_{12} & \cdots & v_{1m} \\ v_{21} & v_{22} & \cdots & v_{2m} \\ \vdots & \vdots & & \vdots \\ v_{m1} & v_{m2} & \cdots & v_{mm} \end{bmatrix}$ and $T = \begin{bmatrix} t_{11} & t_{12} & \cdots & t_{1n} \\ t_{21} & t_{22} & \cdots & t_{2n} \\ \vdots & \vdots & & \vdots \\ t_{m1} & t_{m2} & \cdots & t_{mn} \end{bmatrix}$ represent the spatial function

matrix and time function matrix, respectively. The spatial function matrix extracted from the raw matrix comprises important spatial information and generally has clear physical meanings.

Pearson correlation analysis was used to show the relationship between the NDVI over the Tibetan Plateau and the NVDI over Siberia. To remove the impact of the decadal trend on this relationship, the correlation was calculated using linearly detrended NDVIs over the two regions. Regression analysis was also used. The statistical significance of the correlation coefficients was determined by Student's $t$ test unless stated otherwise.

After finding an atmospheric circulation pattern responsible for the co-variability of the NDVIs, we used the method of stepwise regression [53] to construct an empirical statistical model, which further reveals the influence of the synergistic SSTAs in different oceans on this pattern. The optimal influencing factors that are independent of each other effectively are selected using this method [53].

## Results

### Co-variability of the NDVI between the eastern Tibetan Plateau and Lake Baikal

The climatological (1982–2014) mean summer NDVI (Fig 1A) shows that NDVI over the Tibetan Plateau decreases progressively from southeast to northwest, and the NDVI over the eastern Tibetan Plateau is significantly higher than that over the western Tibetan Plateau, indicating a greater coverage and more active growth of vegetation in the east of this region. The difference of the NDVIs between the eastern and western Tibetan Plateau may be partly due to the fact that the dominant types of vegetation over the eastern Tibetan Plateau are also different from those over the western Tibetan Plateau. Specifically, the eastern Tibetan Plateau is primarily covered by coniferous and broadleaf forests, whereas the western Tibetan Plateau is mainly covered by grassland [35].

This distribution of the NDVIs (Fig 1A) is also in good agreement with the that of the precipitation (Fig 1B). Summer precipitation over the Tibetan Plateau decreases progressively from southeast to northwest, and higher (lower) summer precipitation occurs over the eastern (western) Tibetan Plateau (Fig 1B), which is consistent with the previous result [54]. Water has a crucial role in various physiological processes in plants, such as nutrient transport, the maintenance of structure and the availability of soil water [55]. An appropriate amount of precipitation clearly favors the growth of vegetation. A comparison of the distributions of the NDVI and precipitation over the Tibetan Plateau (Fig 1) shows that the higher precipitation over the eastern Tibetan Plateau favors a higher coverage of vegetation and *in situ* growth activity. The eastern Tibetan Plateau is more significantly affected by climate change than other regions of the Tibetan Plateau [56] and therefore this study focused on the variability of the NDVI on the eastern Tibetan Plateau.

For convenience, the regional mean NDVI over the eastern Tibetan Plateau (28–35˚ N, 99˚–104˚ E) is defined as the eastern Tibetan Plateau NDVI index (hereinafter called the Tibetan Plateau NDVI index). Fig 2A shows the correlation between the summer Tibetan Plateau NDVI index and the simultaneous NDVI field during the time period 1982–2014. The variation in the NDVI is not consistent over the whole Tibetan Plateau region—for example, the NDVI on the eastern Tibetan Plateau is not significantly correlated with that on the western Tibetan Plateau, revealing that the variation of the NDVI on the eastern Tibetan Plateau has its own local characteristics. Interestingly, a significant correlation appears over Lake Baikal and the region to the east, suggesting that there might be a teleconnection between the NDVI over the eastern Tibetan Plateau and that over the Lake Baikal region, showing a co-variability pattern (Fig 2A).

To further verify this co-variability, the regional mean NDVI over the Lake Baikal region (50–59˚ N, 104–126˚ E; upper green box in Fig 2A) was defined as the Lake Baikal NDVI index and the correlation between the summer Lake Baikal NDVI index and the simultaneous NDVI field during the time period 1982–2014 was calculated (Fig 2B). Fig 2B also clearly shows an in-phase variation of the NDVI over the Lake Baikal and eastern Tibetan Plateau regions, reflecting a co-variability pattern (Fig 2A). The time series of the summer Tibetan Plateau and Lake Baikal NDVI indices (Fig 3A) further show the consistent variability of the NDVI between the two regions with a correlation coefficient of 0.62 (significant at the 99% confidence level).

The NDVI over the Lake Baikal region shows a weak increasing trend of 0.11/decade (non-significant), which is generally consistent with previous studies [57], although the specific values of trends are different because of selecting different regions in Siberia. The normalized NDVI over the eastern Tibetan Plateau shows a decreasing trend of −0.33/decade (significant

(a) NDVI

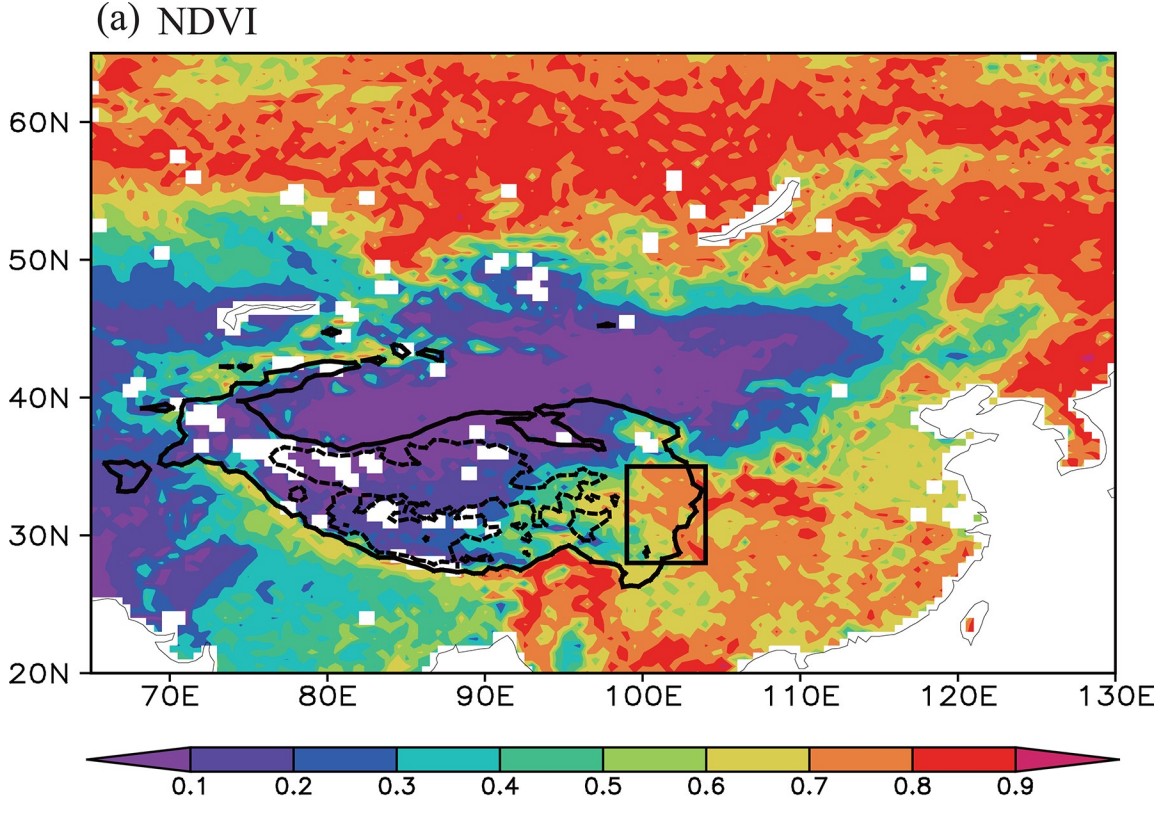

(b) Precipitation

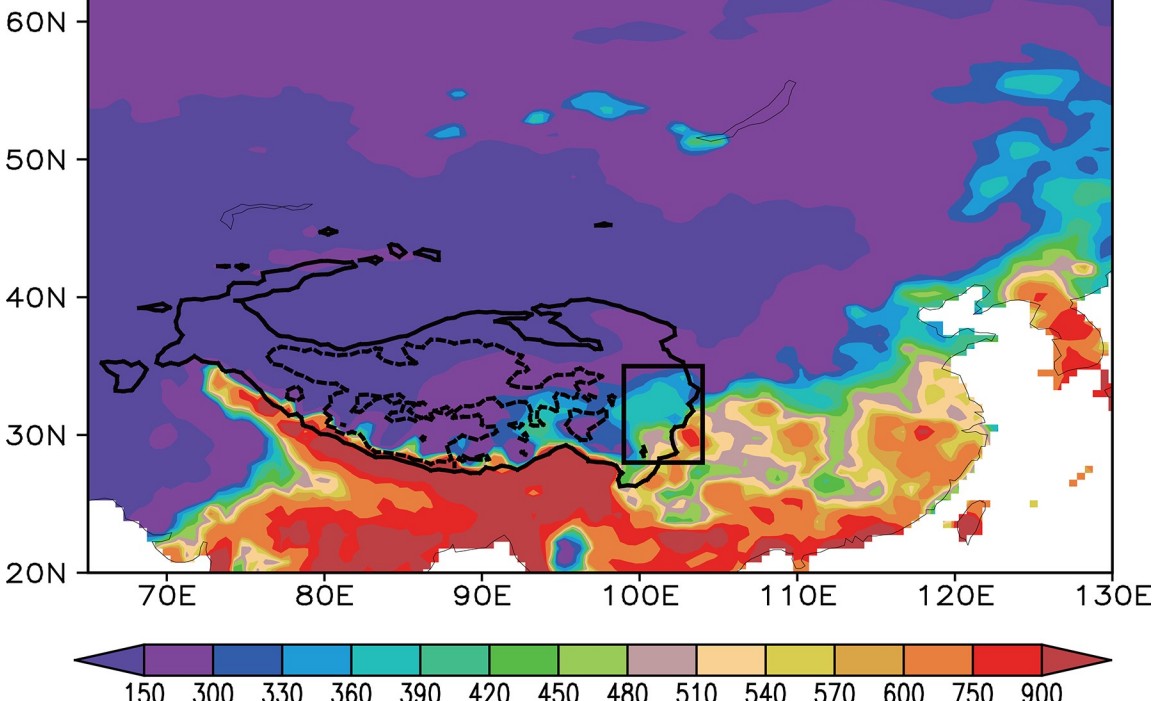

**Fig 1.** Climatological mean summer (a) NDVI and (b) precipitation (units: mm) for the time period 1982–2014. The solid and dashed contours denote areas above 3000 and 5000 m altitude over the Tibetan Plateau. The black box (28–35˚ N, 99–104˚ E) represents the eastern Tibetan Plateau region where there is higher precipitation and a higher NDVI relative to other regions of the Tibetan Plateau.

at the 90% confidence level), whereas the normalized NDVI over the central and western Tibetan Plateau generally shows an increasing trend (figure omitted). This difference may be attributable to different vegetation types, which lead to the spatial heterogeneity of the response of the NDVI to climate change over the Tibetan Plateau [2,7].

To eliminate the influence of the decadal trends in the NDVIs, we recalculated the correlation between the Tibetan Plateau/Lake Baikal NDVI index and simultaneous NDVI field after removing the linear trends from all these variables (Fig 2C and 2D). A similar co-variability appears in the NDVIs for the eastern Tibetan Plateau and the Lake Baikal region (Fig 2C and 2D). The detrended Tibetan Plateau and Lake Baikal NDVI indices clearly show a consistent variation (Fig 3B) with a correlation coefficient of 0.69 (significant at the 99% confidence level), which is higher than that between the raw NDVI indices.

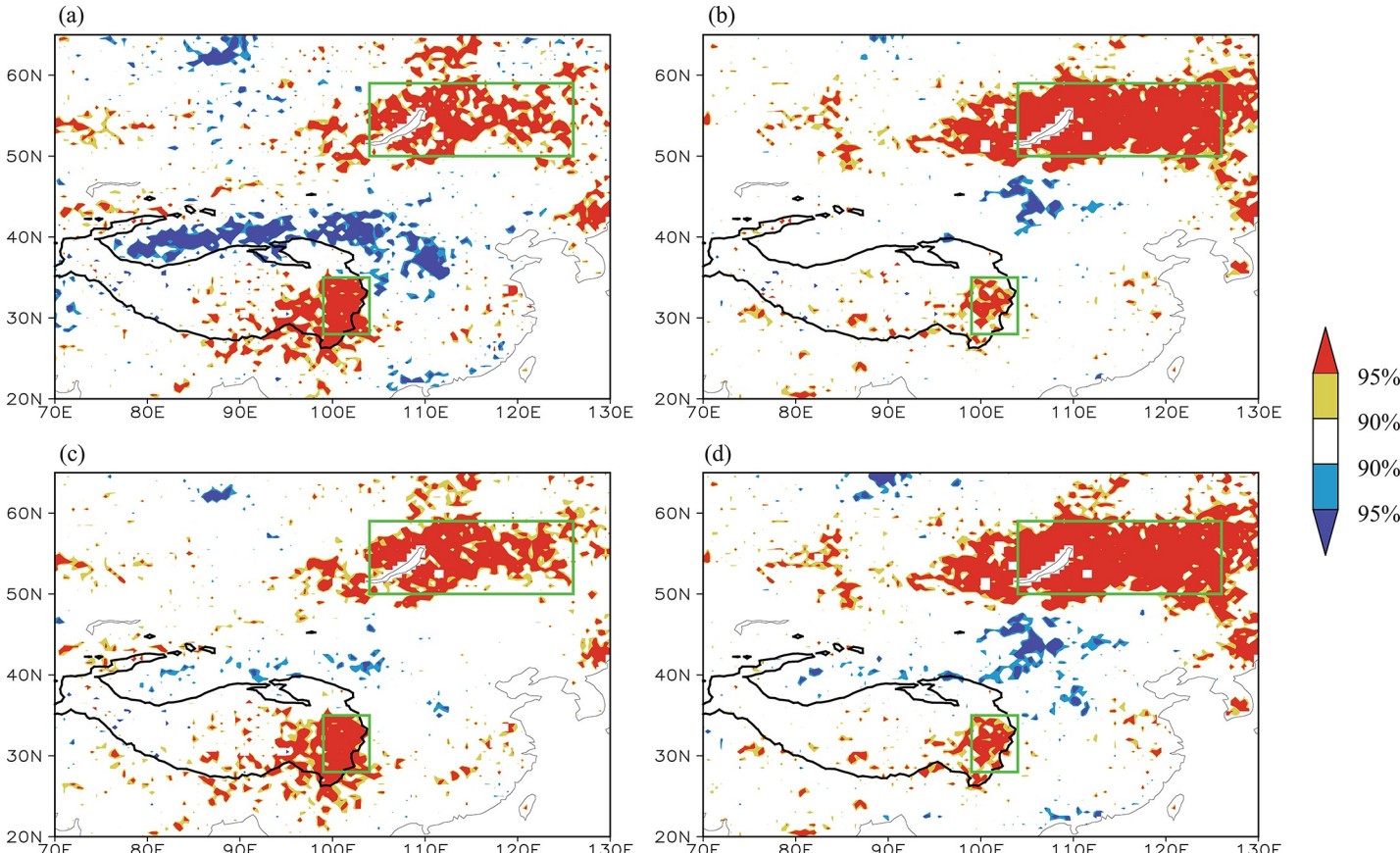

**Fig 2.** (a) Distribution of the correlation coefficients between the summer NDVI index on the Tibetan Plateau and the simultaneous NDVI field during the time period 1982–2014. (b) Distribution of the correlation coefficients between the summer NDVI index in the Lake Baikal region and the simultaneous NDVI field. (c) Distribution of the correlation coefficients between the summer NDVI index on the Tibetan Plateau and the simultaneous NDVI field during the time period 1982–2014 after removing the linear trends. (d) Distribution of the correlation coefficients between the summer NDVI index in the Lake Baikal region and the simultaneous NDVI field during the time period 1982–2014 after removing the linear trends. The solid contour denotes areas higher than 3000 m altitude. The yellow (red) shading denotes a positive correlation significant at the 90% (95%) confidence level and the blue (purple) shading denotes a negative correlation significant at the 90% (95%) confidence level. The two green boxes from top to bottom indicate the Lake Baikal (50–59˚ N, 104–126˚ E) and eastern Tibetan Plateau regions, respectively.

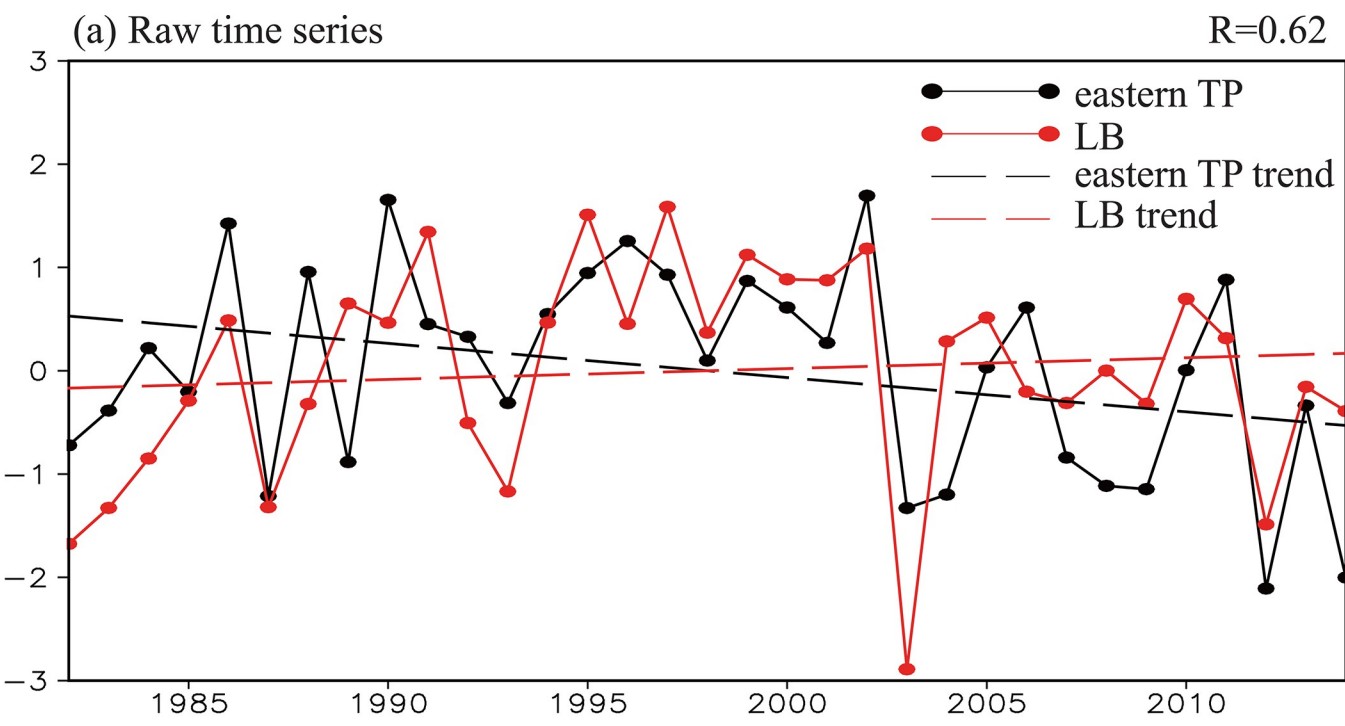

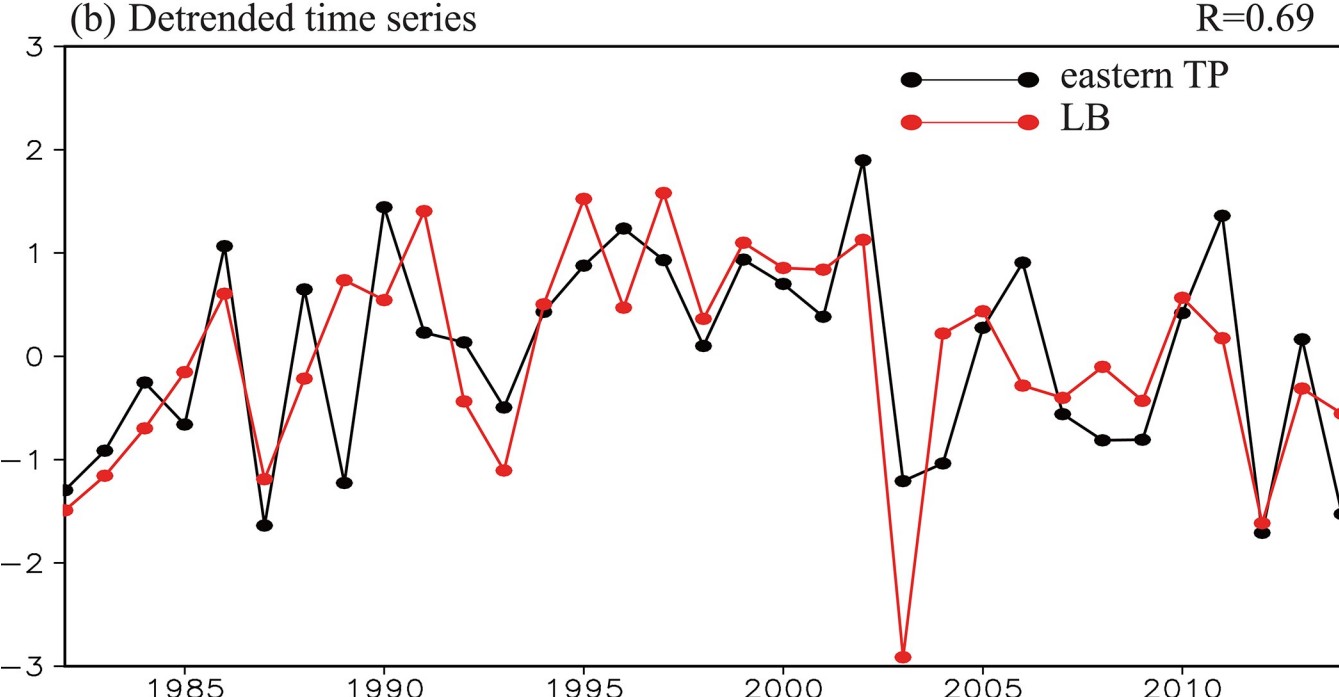

**Fig 3.** (a) Normalized time series of the summer Tibetan Plateau (black line) and Lake Baikal (red line) NDVI indices during the period 1982–2014. The black and red dashed lines denote the linear trends of the Tibetan Plateau and Lake Baikal NDVI indices, respectively. (b) Normalized time series of the summer Tibetan Plateau (black line) and Lake Baikal (red line) NDVI indices during the period 1982–2014 for the detrended time series. The black and red dashed lines denote the linear trends of the Tibetan Plateau and Lake Baikal NDVI indices, respectively. The correlation coefficient is 0.62 for the two raw NDVI indices and 0.69 for the detrended indices, both significant at the 99% confidence level. LB, Lake Baikal; TP, Tibetan Plateau.

These results show that there is a close relationship between the NDVI over the eastern Tibetan Plateau and the NVDI over the Lake Baikal region with a clear, in-phase co-variability. This co-variability is more significant on an interannual timescale. All the datasets discussed in the following sections are detrended.

## Possible reasons for the co-variability of the NDVI

**Climatic factors affecting the two regions.** Before analyzing the causes of the co-variability of the NDVI between the eastern Tibetan Plateau and Lake Baikal regions, we investigated the climate factors governing the variations in the NDVI over the two regions. A number of studies have reported that temperature and precipitation play important parts in regulating the NDVI in the Tibetan Plateau region [2,7,9–11]. Du et al. [6] found that the NDVI on the Tibetan Plateau is closely related to the duration of sunshine. However, the response of the NDVI to climate factors in different regions of the Tibetan Plateau are not the same and show a clear temporal and spatial heterogeneity [2,7,16,17]. To determine the climate factors affecting the NDVI over the eastern Tibetan Plateau, we calculated the correlation coefficients between the summer Tibetan Plateau NDVI index and the simultaneous regionally averaged temperature/precipitation over the eastern Tibetan Plateau during the time period 1982–2014 (Table 1). The results show that the NDVI over the eastern Tibetan Plateau is significantly and negatively correlated with the local precipitation with a correlation coefficient of –0.46 (significant at the 99% confidence level), but is not significantly linked with the local surface air temperature (Table 1).

The NDVI and precipitation over the eastern Tibetan Plateau show a negative rather than a positive correlation on an interannual timescale. In terms of the climatological mean, the higher precipitation over the eastern Tibetan Plateau favors a higher coverage of vegetation and higher growth activity. This background precipitation (Fig 1B) can be considered as the most appropriate amount for the growth of vegetation on the eastern Tibetan Plateau. When the amount of precipitation reaches or exceeds this appropriate amount (or critical value), it will no longer be the driving factor for the growth of vegetation [58]. A greater amount of precipitation may increase the soil moisture content and the latent heat of evaporation, which is not conducive to photosynthesis and therefore may subsequently inhibit the activity of vegetation [55,59,60].

A greater amount of precipitation also corresponds to more cloud cover and less solar radiation, leading to a reduction in photosynthesis and associated plant growth. The Tibetan Plateau NDVI index shows a significant positive correlation with the simultaneous regional mean OLR and solar radiation indices over the eastern Tibetan Plateau (Table 1), supporting the suggestion that less (more) cloud cover and more (less) solar radiation favor a higher (lower) NDVI over the eastern Tibetan Plateau. This suggests that higher precipitation (more cloud cover/less solar radiation) leads to a lower NDVI over eastern Tibetan Plateau on an interannual timescale, showing a significant anticorrelation. This precipitation–NDVI anticorrelation is roughly consistent with previous studies in different regions [6,61].

**Table 1. Correlation coefficients between the Tibetan Plateau–Lake Baikal NDVI index and local climate factors (precipitation, surface air temperature, OLR and solar radiation) on an interannual timescale.**

|  | Precipitation | Surface air temperature | OLR | Solar radiation |
|---|---|---|---|---|
| **Tibetan Plateau NDVI** | −0.46** | 0.16 | 0.54** | 0.43* |
| **Lake Baikal NDVI** | −0.25 | 0.46** | 0.44* | 0.21 |

* and ** denote that the correlation coefficient is significant at the 95 and 99% confidence level, respectively.

Correlation analyses show that the surface air temperature is the most important factor affecting the NDVI over the Lake Baikal region. This can be clearly identified in Table 1, where the correlation coefficient between the NDVI and the surface air temperature is the highest (0.46, significant at the 99% confidence level). The result is consistent with previous studies [23,62–64].

To sum up, precipitation and the associated cloud cover and solar radiation may be the key factors affecting the NDVI over eastern Tibetan Plateau on an interannual timescale, whereas the surface air temperature has a greater role in adjusting the NDVI over the Lake Baikal region. Although different climate factors contribute to the variability of the NDVI over the eastern Tibetan Plateau and Lake Baikal regions, there may be a specific atmospheric circulation anomaly that concurrently modulates these climate factors over the two regions and hence causes the co-variability of the NDVI.

**Atmospheric circulation anomalies responsible for the co-variability of the NDVI.** Fig 4A shows the distribution of the correlation coefficients between the summer Tibetan Plateau NDVI index and the simultaneous 200 hPa geopotential height field during the time period 1982–2014. There is a significant negative correlation from the southeastern Tibetan Plateau to southern China and a significant positive correlation from the north of the Tibetan Plateau to the Lake Baikal region, forming a clear north–south dipole pattern. The correlation between the summer Lake Baikal NDVI index and the simultaneous 200 hPa geopotential height field (Fig 4B) also displays a dipole pattern resembling Fig 4A. This similarity suggests that the NDVIs over the eastern Tibetan Plateau and Lake Baikal regions are regulated by the same atmospheric circulation pattern, namely, a dipole pattern.

If this dipole pattern appears often and is a dominant pattern of the atmospheric circulation anomalies, it could cause the co-variability of NDVI between the eastern Tibetan Plateau and Lake Baikal regions. To confirm this speculation, the dominant modes of the summer 200 hPa

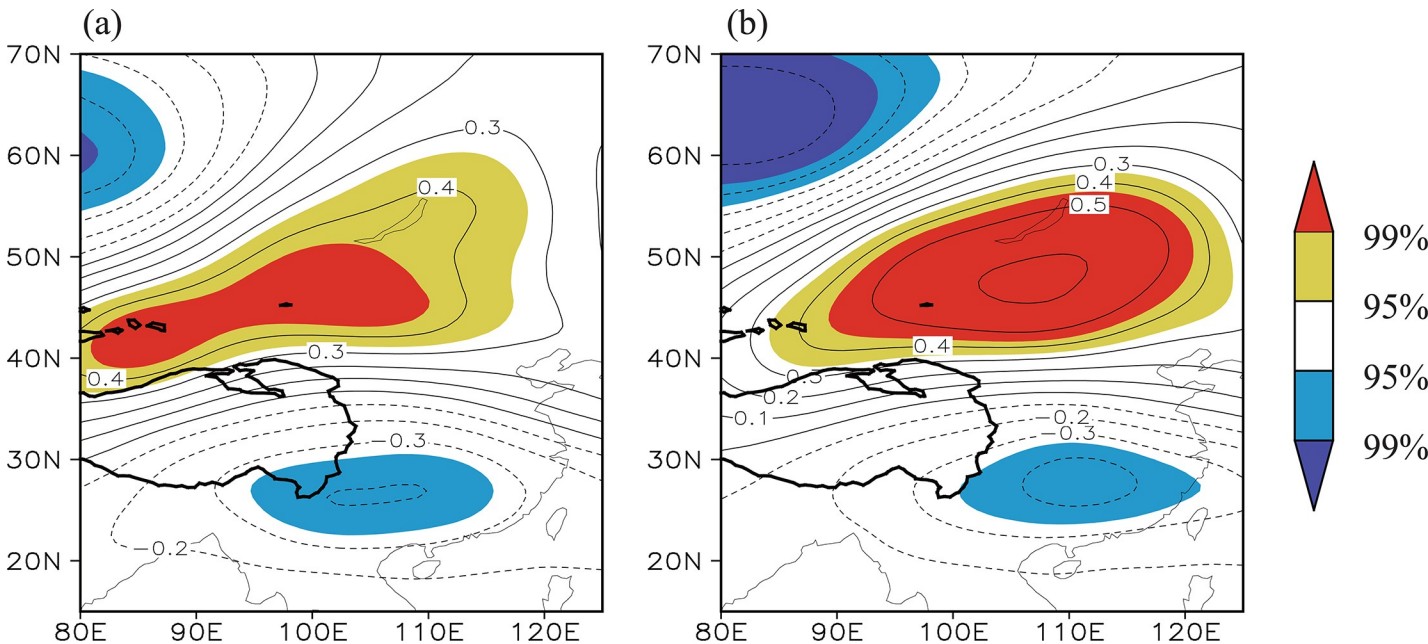

**Fig 4.** (a) Distribution of the correlation coefficients between the Tibetan Plateau NDVI index and the simultaneous 200 hPa geopotential height field during the time period 1982–2014. (b) Distribution of the correlation coefficients between the Lake Baikal NDVI index and the geopotential height field. The solid contour denotes areas higher than 3000 m altitude. The yellow (red) shading denotes a positive correlation significant at the 95% (99%) confidence level and the blue (purple) shading denotes a negative correlation significant at the 95% (99%) confidence level.

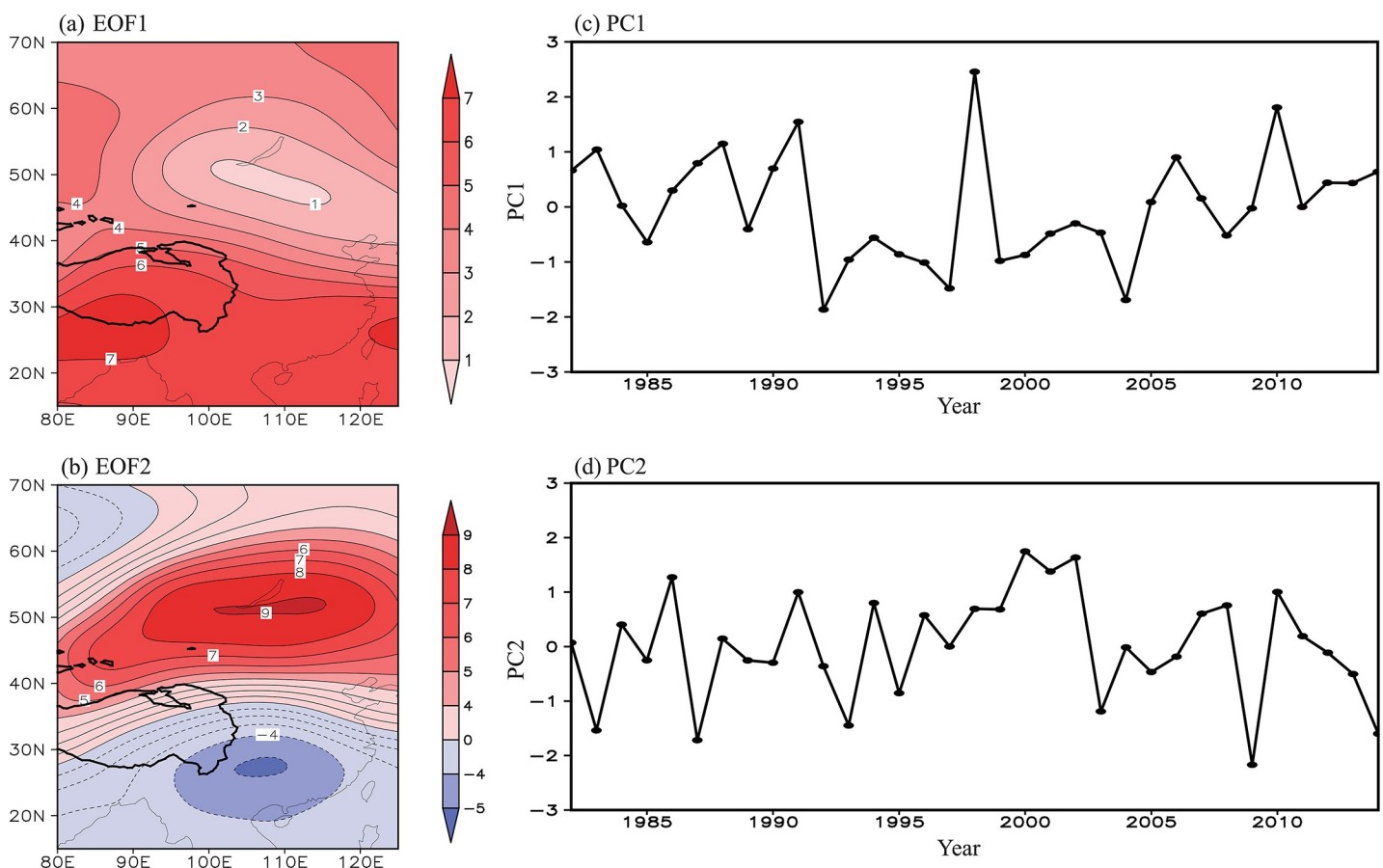

**Fig 5.** The (a) first (EOF1) and (b) second (EOF2) EOF mode for the normalized summer 200 hPa geopotential height field and the time series of the associated (c) PC1 and (d) PC2 during the time period 1982–2014. The explained variance of EOF1 (EOF2) is 36.7% (21.8%). The values of EOF1 and EOF2 are multiplied by 100.

geopotential height field were extracted using the EOF method. The first EOF mode (EOF1) is characterized by an in-phase variation over the full range, with a center of loading to the south of the Tibetan Plateau (Fig 5A). This mode accounts for 36.7% of the total variance. The second EOF mode (EOF2), which accounts for 21.8% of the total variance, is characterized by an out-of-phase variation in the geopotential height over East Asia, showing a north–south dipole pattern similar to Fig 4 (Fig 5B). This verifies that the dipole pattern is one of the most important patterns in the atmospheric circulation anomalies, governing a considerable part of the variability in the atmospheric circulation over this region. The correlation coefficient between the time series of the principal component (PC2; Fig 5D) of EOF2 and the Tibetan Plateau/ Lake Baikal NDVI indices is 0.55/0.55 (significant at the 99% confidence level). This further supports that this dipole pattern, measured by PC2 (Fig 5D), is synchronously linked with the NDVIs over the eastern Tibetan Plateau and Lake Baikal regions. As such, PC2 is referred to here as the Tibetan Plateau–Lake Baikal dipole index.

Fig 6A shows the climatological (1982–2014) mean summer eddy 200 hPa geopotential height field, in which the eddy geopotential height is defined as the deviation of the geopotential height from the zonal mean at the same latitude. This eddy geopotential height can clearly reflect high-/low-pressure anomalies along the same latitude [65]. Fig 6A clearly shows a large high-pressure zone ranging from the Arabian Peninsula to southern China via the Tibetan Plateau. This high-pressure zone is known as the South Asian high, the center of which is a 100

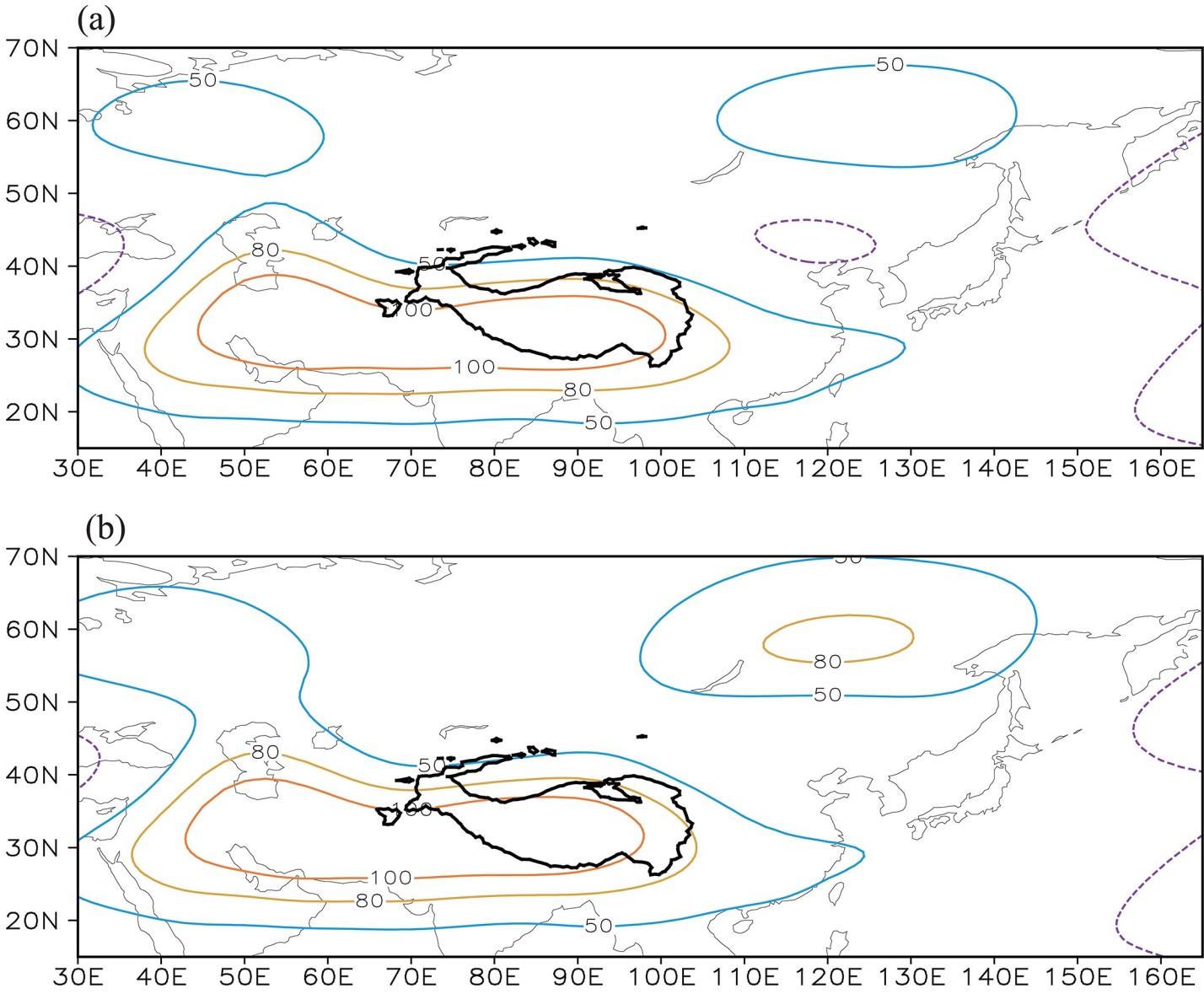

**Fig 6.** (a) Spatial distribution of the climatological mean summer eddy 200 hPa geopotential height (units: gpm) during the time period 1982–2014. (b) Spatial distribution of the climatological eddy 200 hPa geopotential height after superimposition by the geopotential height anomalies regressed on the Tibetan Plateau–Lake Baikal dipole index (i.e., the principal component of EOF2). The black solid contour denotes the Tibetan Plateau (>3000 m).

gpm high-pressure anomaly extending eastward to the southeastern Tibetan Plateau (Fig 6A). A high-pressure system also appears to the east of the Lake Baikal, centered around (60° N, 120° E), which represents the northeast Asian blocking high.

We calculated the 200 hPa geopotential height anomalies regressed upon the dipole index (i.e., PC2). And then, the geopotential height anomalies were superimposed on the climatological mean 200 hPa eddy geopotential height field (Fig 6B). Through this superimposition, we can intuitively see anomalous characteristics of atmospheric circulation systems associated with this dipole. As shown in Fig 6B, the eastern edge of the South Asian high reflected by the 100 gpm contour withdraws further west than the location in Fig 6A and the northeastern

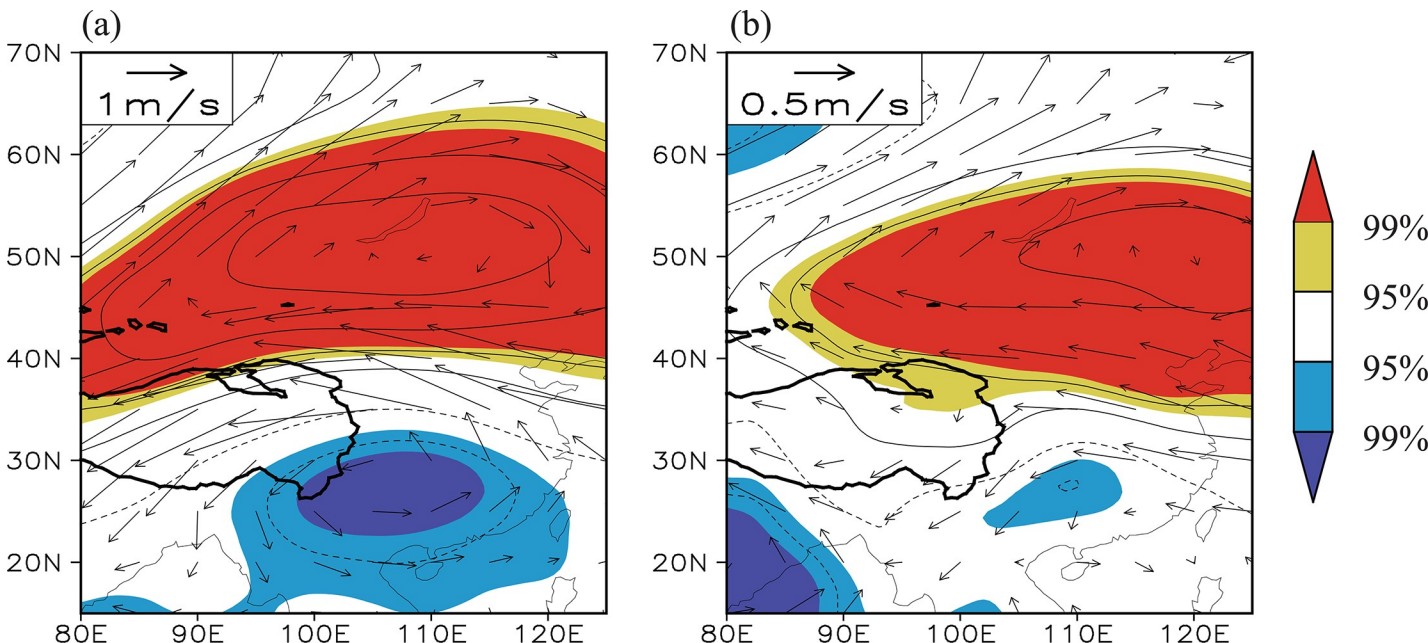

**Fig 7.** (a) Distribution of the correlation coefficients between the Tibetan Plateau–Lake Baikal dipole index and the simultaneous 200 hPa geopotential height field, in which the 200 hPa wind anomalies regressed against the Tibetan Plateau–Lake Baikal dipole index are represented by black vectors. (b) Distribution of the correlation coefficients between the Tibetan Plateau–Lake Baikal dipole index and the simultaneous 600 hPa geopotential height field, in which the 600 hPa wind anomalies regressed against the Tibetan Plateau–Lake Baikal dipole index are represented by black vectors. The solid contour denotes areas higher than 3000 m altitude. The yellow (red) shading denotes a positive correlation significant at the 95% (99%) confidence level and the blue (purple) shading represents a negative correlation significant at the 95% (99%) confidence level.

Asian blocking high intensifies and expands southwestward and controls the Lake Baikal region.

The upper tropospheric (200 hPa) geopotential height over the southeastern Tibetan Plateau often shows an overall out-of-phase variation with that over the Lake Baikal region during summer, indicating a north–south dipole pattern. The Tibetan Plateau–Lake Baikal dipole pattern is the second mode of the atmospheric circulation anomalies in this region and should be regarded as a typical and dominant mode that often appears. In fact, this dipole pattern effectively reflects the anomalies in the strength and location of the South Asian high and the northeast Asian blocking high. An anomalous South Asian high, together with an anomalous northeast Asian blocking high, may modulate climate factors and the related NDVIs over the eastern Tibetan Plateau and Lake Baikal regions and consequently lead to the co-variability of the NDVI between the two regions.

We analyzed how the Tibetan Plateau–Lake Baikal dipole pattern affects different climate factors over the eastern Tibetan Plateau and Lake Baikal regions. The Asian monsoon-induced water vapor transport can modulate precipitation over the Tibetan Plateau [54,66]. Specifically, the southwest monsoon transports water vapor from the Indian Ocean and the South China Sea to the eastern Tibetan Plateau, where it is the main source of water vapor for precipitation in this region [67]. The correlation between the summer Tibetan Plateau–Lake Baikal dipole index and the simultaneous 200 and 600 hPa geopotential height field (Fig 7) shows that the out-of-phase dipole pattern appears in both the upper and mid- to lower troposphere, forming a deep configuration of high-/low-pressure anomalies. Corresponding to a higher Tibetan Plateau–Lake Baikal dipole index, a significant negative geopotential height anomaly appears over southern China (i.e., the westward withdrawal of the South Asian high) and hence an

anomalous cyclone also occurs in this region (vectors in Fig 7). An anomalous northerly movement prevails along the western flank of this anomalous cyclone, which weakens the southwest monsoon and the associated transport of water vapor to the eastern Tibetan Plateau and ultimately decreases precipitation over the eastern Tibetan Plateau. By contrast, the lower Tibetan Plateau–Lake Baikal dipole index corresponds to strengthening of the southwest monsoon and higher precipitation over the eastern Tibetan Plateau. As a result, a significant negative correlation between the Tibetan Plateau–Lake Baikal dipole index and the precipitation field appears over the eastern Tibetan Plateau (Fig 8A)—that is, the Tibetan Plateau–Lake Baikal dipole pattern modulates the variability of the eastern Tibetan Plateau NDVI by adjusting the South Asian high, the southwest monsoon and the associated precipitation over the eastern Tibetan Plateau.

Fig 7 also shows that, corresponding to the higher (lower) Tibetan Plateau–Lake Baikal dipole index, a deep high- (low-)pressure anomaly appears from the northern Tibetan Plateau to the Lake Baikal region (i.e., the anomalous northeastern Asian blocking high), which is conducive to less (more) cloud cover and more (less) solar radiation reaching the Earth's surface and therefore results in higher (lower) temperatures over the Lake Baikal region. This mechanism can successfully explain the significant positive correlation between the Tibetan Plateau–Lake Baikal dipole index and the surface air temperature over the Lake Baikal region (Fig 8B). The Tibetan Plateau–Lake Baikal dipole pattern therefore has an important role in the variability of the Lake Baikal NDVI by regulating the northeastern Asian blocking high.

The Tibetan Plateau–Lake Baikal dipole pattern in geopotential height can therefore simultaneously affect precipitation over the eastern Tibetan Plateau and the temperature over the Lake Baikal region, which are the dominant climate factors driving the variations in the NDVI over the two regions. The Tibetan Plateau–Lake Baikal dipole pattern provides the background of large-scale circulation anomalies for the co-variability of the NDVI between the two regions and should therefore be considered as the key atmospheric circulation pattern.

**Effect of SSTs.**   The SSTA is an important external forcing factor driving anomalies in the atmospheric circulation. Many studies have pointed out that the El Niño/La Niña pattern of SSTAs in the Pacific Ocean [68–72], the SSTAs in the north Pacific Ocean [73,74], the SSTAs in the tropical Indian Ocean [75] and the SSTAs in the north Atlantic Ocean [73] have a great effect on the atmospheric circulation and climate change in East Asia. These studies prompted us to further investigate whether the Tibetan Plateau–Lake Baikal dipole pattern is related to SSTAs.

Fig 9A shows the distribution of correlation coefficients between the summer Tibetan Plateau–Lake Baikal dipole index and the simultaneous SST field during the time period 1982–2014. Significant negative correlations appear in the eastern tropical Pacific Ocean, the subtropical north Atlantic Ocean and the north Indian Ocean, while a significant positive correlation appears in the northwest Pacific Ocean (Fig 9A).

We further explore the optimum composited SSTAs affecting the Tibetan Plateau–Lake Baikal dipole pattern using the method of stepwise regression. The result shows that the synergistic SSTAs in the tropical eastern Pacific Ocean (0–15˚ N, 80–100˚ W), the subtropical north Atlantic Ocean (20–30˚ N, 10–50˚ W) and the northwest Pacific Ocean (28–38˚ N, 136–156˚ E) have the best effectiveness in fitting the Tibetan Plateau–Lake Baikal dipole pattern. The model is established as follows:

$$I_{Dipole} = 0.429 I_{NW-Pacific} - 0.367 I_{SN-Atlantic} - 0.283 I_{TE-Pacific} \qquad (4)$$

in which $I_{Dipole}$ is the Tibetan Plateau–Lake Baikal dipole, and $I_{NW-Pacific}$, $I_{SN-Atlantic}$, and $I_{TE-Pacific}$ represent the SSTs regionally averaged over the northwest Pacific Ocean, the subtropical

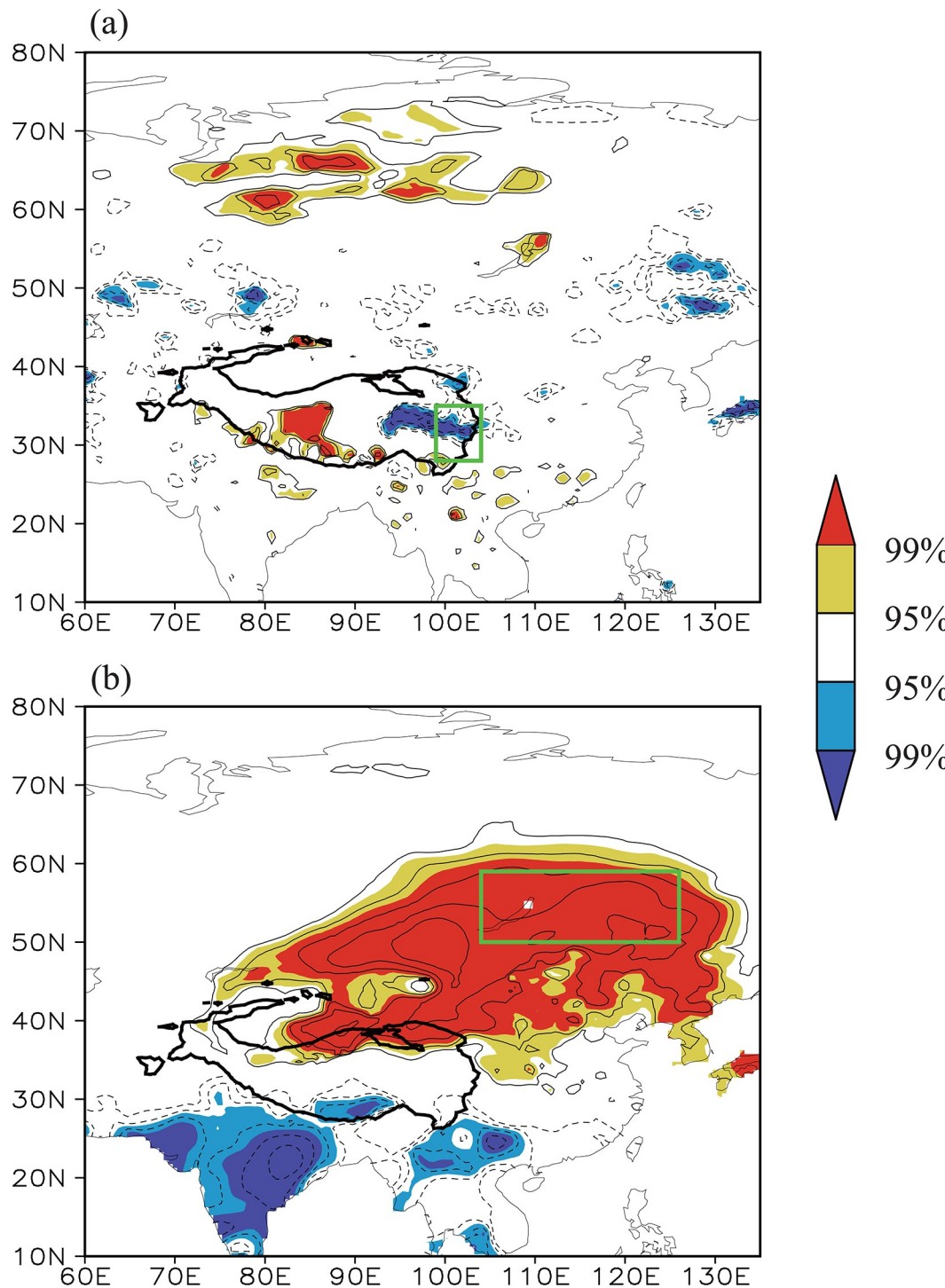

**Fig 8.** Distribution of the correlation coefficients between the Tibetan Plateau–Lake Baikal dipole index and the simultaneous (a) precipitation (b) surface air temperature field during the time period 1982–2014. The solid contour denotes areas higher than 3000 m altitude. The isoline interval is 0.1. The yellow (red) shading denotes a positive correlation significant at the 95% (99%) confidence level and the blue (purple) shading denotes a negative correlation significant at the 95% (99%) confidence level. The two green boxes from top to bottom indicate the eastern Tibetan Plateau and Lake Baikal regions, respectively.

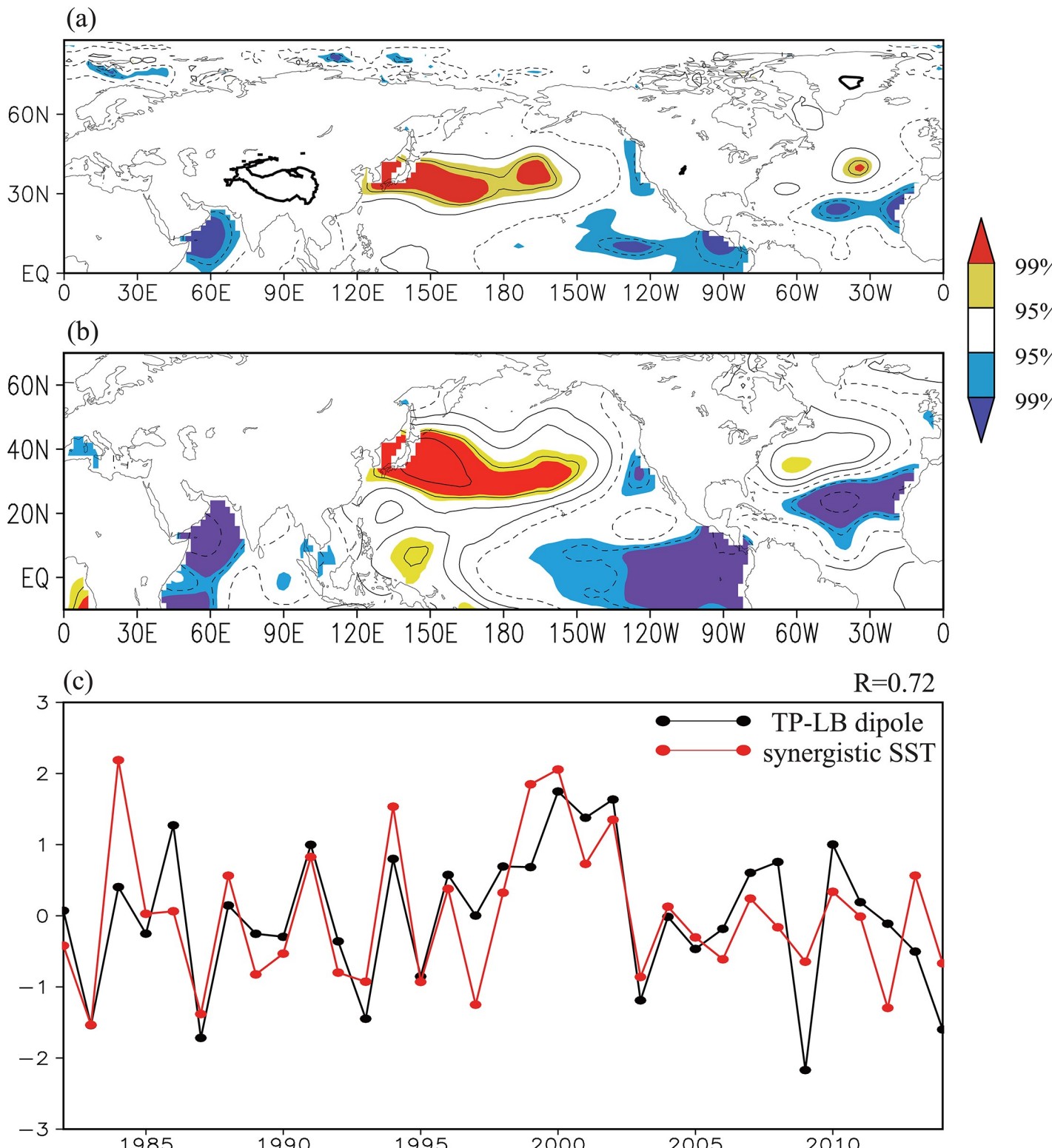

**Fig 9.** (a) Distribution of the correlation coefficients between the Tibetan Plateau–Lake Baikal dipole index and the simultaneous SST field during the time period 1982–2014. (b) as in (a), but for that between the synergistic SST index and the SST field. The solid contour denotes areas higher than 3000 m altitude. The yellow (red) shading denotes a positive correlation significant at the 95% (99%) confidence level and the blue (purple) shading negative correlation significant at the 95% (99%) confidence level. (c) Time series of the normalized Tibetan Plateau–Lake Baikal dipole (black line) and synergistic SST (red line) indices during the time period 1982–2014. TP–LB, Tibetan Plateau–Lake Baikal.

North Atlantic Ocean, and the tropical eastern Pacific Ocean, respectively. The northern Indian Ocean SST index ($I_{NIO}$), which is defined as the SSTs regionally over the northern Indian Ocean (5–25° N, 50–70° E), was removed from this model after stepwise regression processes. All these indices were normalized.

The right terms can reflect the effect of the synergistic SSTAs in these oceans with different weights and therefore are referred to as the synergistic SST index. Fig 9C shows that the variability of the synergistic SST index is in good agreement with that of the Tibetan Plateau–Lake Baikal dipole index, with a high correlation of 0.72 (significant at the 99.9% confidence level). The correlation coefficient between the synergistic SST index and the Tibetan Plateau/Lake Baikal NDVI index is 0.45/0.40 (both significant at the 95% confidence level). These results imply that this configuration of SSTAs seems to regulate the Tibetan Plateau–Lake Baikal dipole pattern of atmospheric circulation and subsequently causes the co-variability of the NDVI between the eastern Tibetan Plateau and Lake Baikal regions.

Next, we explore the potential contributions of SSTAs in different regions to the variability of the Tibetan Plateau–Lake Baikal dipole, respectively. Cooler (warmer) SSTs in the tropical eastern Pacific correspond to anomalous downward (upward) motion in the tropical eastern Pacific and anomalous upward (downward) motion around the western North Pacific through enforcing (restraining) the Pacific Walker circulation [76–78]. As a result of the above process, negative (positive) geopotential height anomalies appear around southern China and the South China Sea in correspondence to negative (positive) SSTAs in the tropical eastern Pacific. This can be detected in the correlation between the summer negative $I_{TE-Pacific}$ and the simultaneous 200 hPa geopotential heights (Fig 10A). For ease of understanding, we used the negative $I_{TE-Pacific}$ to perform correlation analysis since the negative $I_{TE-Pacific}$ corresponds to the positive Tibetan Plateau–Lake Baikal dipole index. As shown in Fig 10A, the SSTAs in the tropical eastern Pacific Ocean may primarily modulate the south pole of the Tibetan Plateau–Lake Baikal dipole.

Note that the correlation between the summer synergistic SST index and the simultaneous SST field (Fig 9B) shows that apart from significant correlations over the northwest Pacific Ocean, the subtropical north Atlantic Ocean, and the tropical eastern Pacific Ocean, a significant negative correlation appears over the northern Indian Ocean. The $I_{NIO}$ is highly correlated with the synergistic SST index, with a correlation coefficient of −0.61 (significant at the 99.9% confidence level). This result implies that the synergistic SST index also reflects the variability of the $I_{NIO}$ though the synergistic SST domain does not cover the northern Indian Ocean. The correlation between the summer negative $I_{NIO}$ and the simultaneous 200 hPa geopotential height field (figure omitted) reveals that negative correlations appear around southern China and the South China Sea, resembling Fig 10A. This suggests that the variability of the SSTs in the northern Indian Ocean may also play an important role in modulating the south part of the Tibetan Plateau–Lake Baikal dipole. The SSTAs in the tropical eastern Pacific Ocean and the northern Indian Ocean seems to affect the south part of this dipole through a pair of gears (strong coupling) between the Indo-monsoon circulation and the Pacific Walker circulation [79–80].

The SSTAs in the tropical Atlantic Ocean can affect downstream climate through stimulating wave train pattern [81–83]. The correlation between the negative $I_{SN-Atlantic}$ between 200 hPa V-winds (Fig 11) shows two wave trains from the Atlantic to East Asia at the mid-low and mid-high latitudes, respectively, in which the wave train at the mid-low latitudes is generally similar to the Silk Road teleconnection [84,85]. Through exciting the wave trains, the SSTAs in the subtropical north Atlantic Ocean result in geopotential height anomalies around the Lake Baikal (Fig 10B)—that is, the $I_{SN-Atlantic}$ may modulate the north part of the Tibetan Plateau–Lake Baikal dipole.

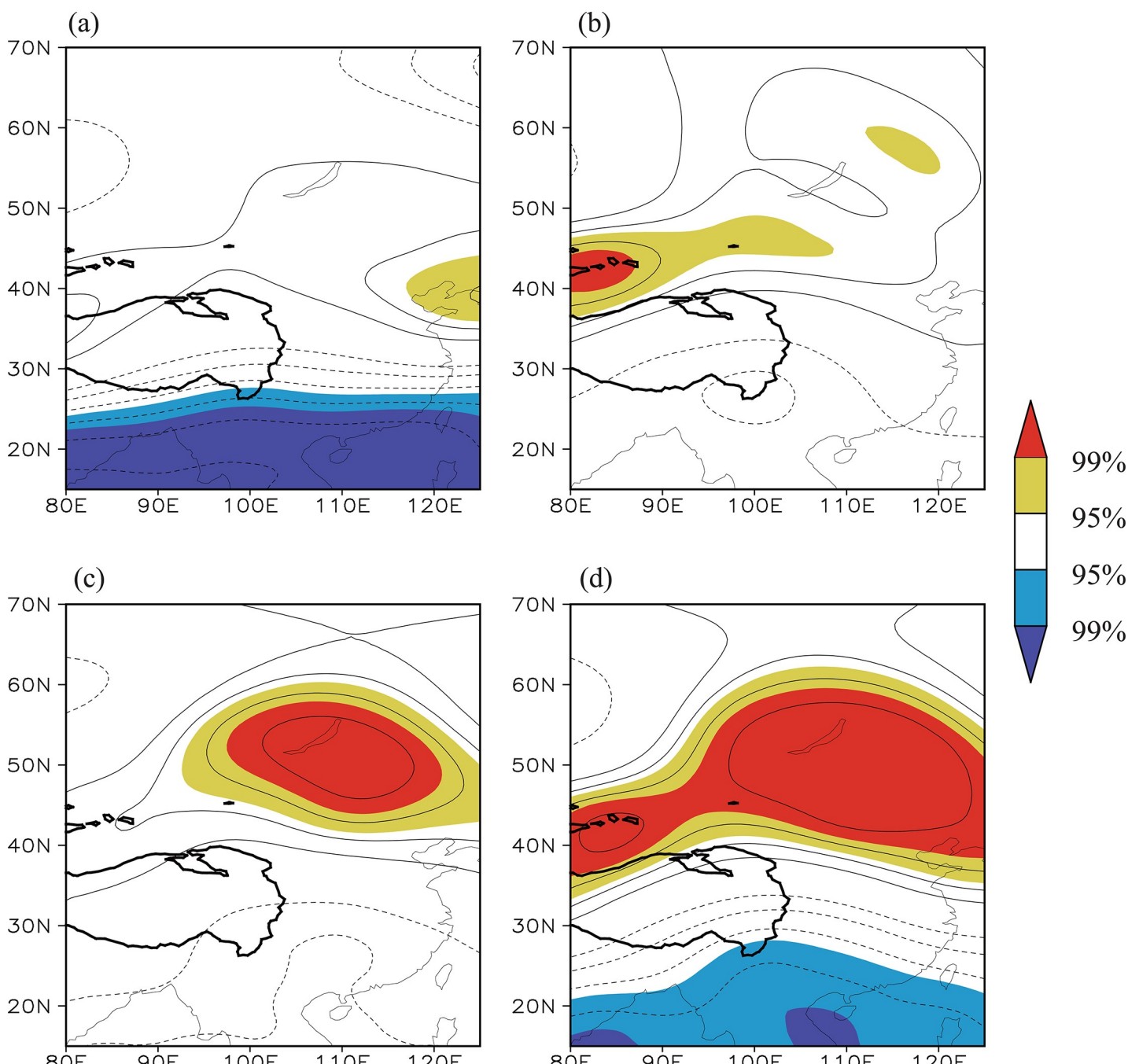

**Fig 10.** Distribution of the correlation coefficients between the summer (a) negative tropical eastern Pacific Ocean SST index/(b) negative subtropical north Atlantic Ocean SST index/(c) positive northwest Pacific Ocean SST index/(d) synergistic SST index and the simultaneous 200 hPa geopotential height field during the time period 1982–2014. The yellow (red) shading denotes a positive correlation significant at the 95% (99%) confidence level and the blue (purple) shading denotes a negative correlation significant at the 95% (99%) confidence level.

The vertical circulation and geopotential height anomalies regressed upon the $I_{NW-Pacific}$ on a cross section along 155˚E (Fig 12A) show anomalous upward motions over the northwest Pacific Ocean and correspondingly a significant positive anomaly appears in the upper troposphere around 40˚ N, to the east of Japan, which can be regarded as a result of a warmer SST

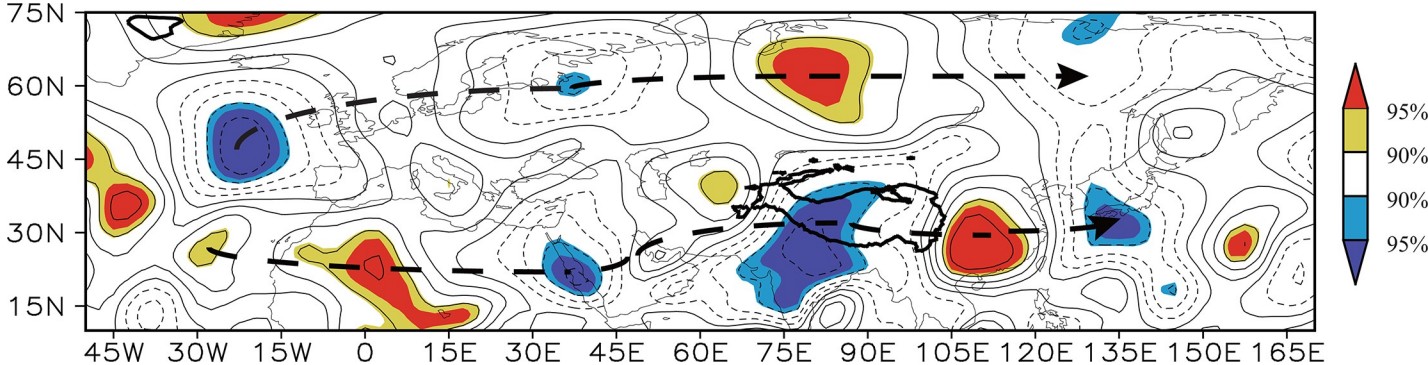

**Fig 11. Distribution of the correlation coefficients between the negative $I_{SN-Atlantic}$ and the simultaneous 200 hPa $V$-wind field during the time period 1982–2014.**
The yellow (red) shading denotes a positive correlation significant at the 90% (95%) confidence level and the blue (purple) shading denotes a negative correlation significant at the 90% (95%) confidence level. The black dashed lines denote the Rossby wave trains.

forcing. The 200 hPa geopotential height anomalies regressed upon the regional mean 200 hPa geopotential height to the east of Japan (32–42˚ N, 145–165˚ E) show a significant positive anomaly around the Lake Baikal (Fig 12B), implying that when a blocking-like high-pressure anomaly appears to the east of Japan, the high-pressure anomalies (i.e., the northeast Asian blocking high) tend to occur around the Lake Baikal. Through forcing the overlying atmospheric circulation anomalies over the northwestern Pacific Ocean, the $I_{NW-Pacific}$ seems to modulate the north part of the Tibetan Plateau–Lake Baikal dipole. Therefore, the $I_{NW-Pacific}$ is significantly correlated with geopotential potential heights around the Lake Baikal (Fig 10C).

In summary, the synergistic SST index effectively reflects a pattern of the SSTAs in the tropical eastern Pacific Ocean, the northern Indian Ocean, the subtropical north Atlantic Ocean, and the northwest Pacific Ocean (Fig 9B). Through the aforementioned physical processes, the SSTAs in different regions seem to affect different parts of the Tibetan Plateau–Lake Baikal dipole and jointly modulate this dipole pattern. Thus, the synergistic SSTA pattern is closely linked with this atmospheric dipole pattern (Fig 10D).

## Discussion

Vegetation growth might be successive during the growing season (May–October for the Tibetan Plateau) [86]. It is therefore necessary to further clarify whether the co-variability of the summer NDVIs between the eastern Tibetan Plateau and Lake Baikal regions is only a result of the maintenance of the NDVI anomalies from spring (April–May) to summer. The NDVI over the Lake Baikal region persists from spring to summer, which can be revealed by the high correlation coefficient (0.43) between the spring and summer Lake Baikal NDVI indices, significant at the 99% confidence level. In contrast, the NDVI over the eastern Tibetan Plateau does not persist, since the correlation coefficient between the spring and summer Tibetan Plateau NDVI indices is only 0.26 (non-significant). There is no co-variability between the NDVI over the eastern Tibetan Plateau and that over the Lake Baikal region in spring (the correlation coefficient is very low, only –0.04). These results show that the co-variability of the NDVI during summer is not the result of the cross-seasonal persistence of NDVIs from spring to summer, but can be attributed to the concurrent regulation of atmospheric circulation systems and the associated climate factors during summer.

According to the sensible heat air pumping effect of the Tibetan Plateau proposed by Wu et al [87,88], anomalous surface sensible heating can drive upward motion and associated

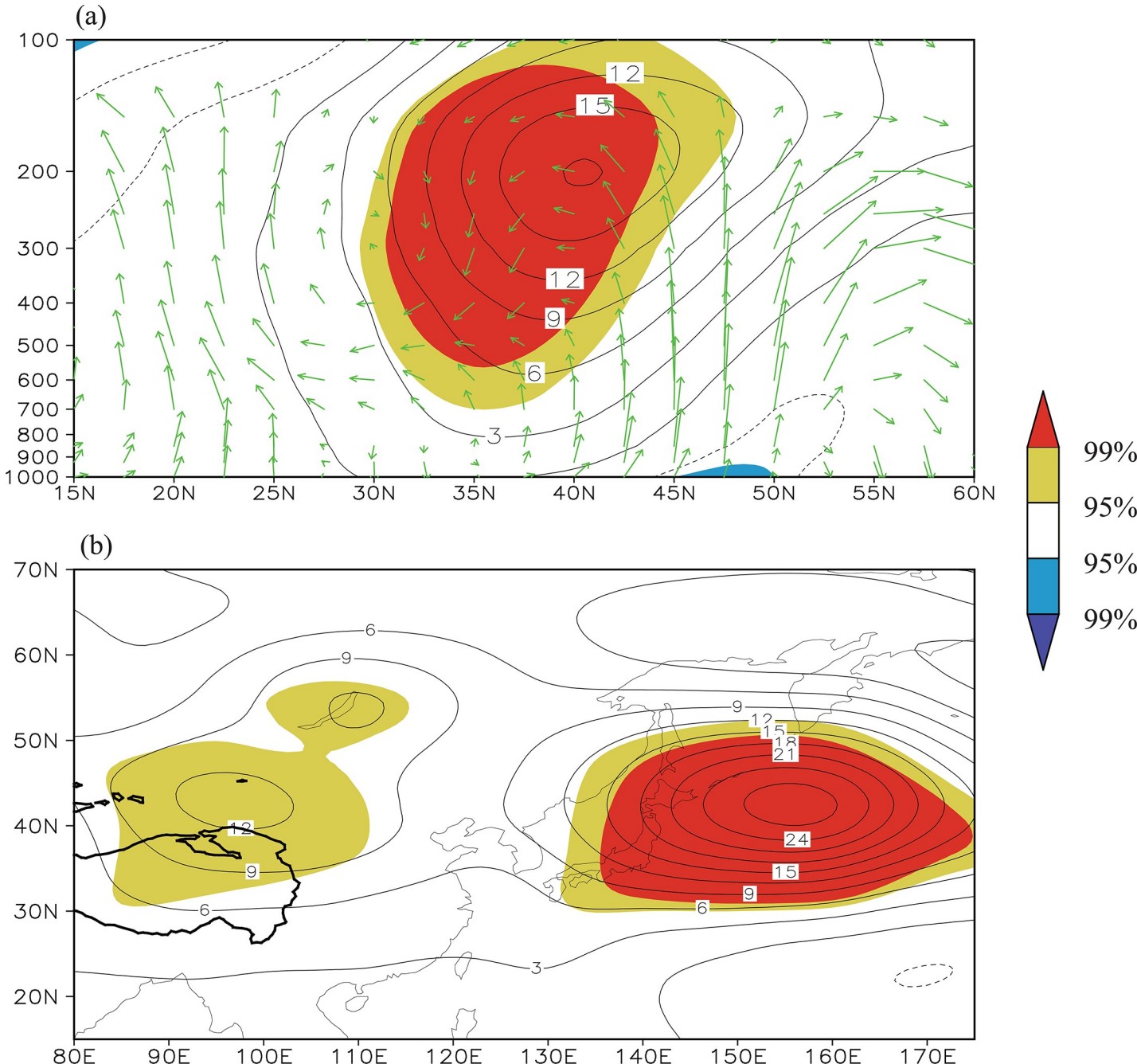

**Fig 12.** (a) Summer vertical motion and geopotential height anomalies regressed upon the northwest Pacific Ocean SST index on a cross section along 155˚E. (b) The 200 hPa geopotential height anomalies regressed on the regional mean 200-hPa geopotential height to the east of Japan (32–42˚ N, 145–165˚ E) during the time period 1982–2014. The yellow (red) shading denotes a positive correlation significant at the 95% (99%) confidence level and the blue (purple) shading denotes a negative correlation significant at the 95% (99%) confidence level.

circulation anomalies over the Tibetan Plateau and therefore exert an important effect on climate change in East Asia [89–91]. This implies a potential feedback effect of the thermal anomalies associated with the NDVI on the atmospheric circulation anomalies over the Tibetan Plateau and adjacent regions. However, further analyses (figure omitted) reveal that the vegetation-related surface heat flux and vertical circulation anomalies are inconsistent with

the theory of sensible heat air pumping and are not the determinant factors of the Tibetan Plateau–Lake Baikal dipole pattern. In turn, this atmospheric circulation pattern modulates the co-variability of the vegetation.

We found a close relationship between the Tibetan Plateau–Lake Baikal dipole pattern and the SSTAs and preliminarily explained the associated physical link between this pattern and the SSTAs in different oceans. Nevertheless, the physical mechanism requires further investigation in detail. For example, it should be further demonstrated why a blocking-like high-pressure anomaly to the east of Japan, which is forced by the SSTAs in the northwest Pacific, is conducive to the high-pressure anomalies around the Lake Baikal. Numerical experiments are also needed to clarify the impact of SSTAs in different oceans in the future.

Although these issues are still unclear, this study shows that there is a co-variability in vegetation between the Tibetan Plateau and Lake Baikal regions and presents a preliminary interpretation of the possible mechanism for this co-variability from the perspective of climate factors, atmospheric circulation anomalies and SSTAs. This may contribute to improve understanding of the characteristics and causes of vegetation co-variability in diverse regions and provide a scientific basis for ecological conservation and construction.

## Conclusions

The vegetation dynamics in the Tibetan Plateau and Siberia are both sensitive to climate change and the characteristics of the NDVIs and the climate factors governing their variation have been widely reported. Nevertheless, few studies have examined the relationship between the NDVI in the Tibetan Plateau and that in Siberia and whether the atmospheric circulation anomalies are responsible for the variations in vegetation. This study focused on the link between the NDVIs of the two regions and tries to explain this link in terms of the climate factors (temperature and precipitation), the background atmospheric circulation and the potential effect of SSTAs. The results are as follows.

1. The summer NDVI over the eastern Tibetan Plateau is significantly and positively correlated with the summer NDVI over the Lake Baikal region in Siberia during the time period 1982–2014, reflecting an in-phase co-variability. This co-variability of the NDVI between the two regions is more significant on an interannual timescale.

2. Precipitation and the related cloud cover and solar radiation are responsible for the variability of the NDVI over the eastern Tibetan Plateau, whereas the temperature plays a more essential part in modulating the variability of the NDVI over the Lake Baikal region.

3. The geopotential height in the upper troposphere over the southeastern Tibetan Plateau and that over the Lake Baikal region show an out-of-phase variation during summer, constituting a Tibetan Plateau–Lake Baikal dipole pattern. This dipole pattern is one of the most important patterns of the atmospheric circulation anomalies over this region. The dipole pattern reflects the anomalies in the intensity and location of the South Asian high and the northeast Asian blocking high, which can concurrently modulate precipitation over the eastern Tibetan Plateau and the surface air temperature over the Lake Baikal region and eventually lead to the co-variability of the NDVI between the eastern Tibetan Plateau and the Lake Baikal regions.

4. A synergistic SST index, which effectively reflects a pattern of the SSTAs in the tropical eastern Pacific Ocean, the northern Indian Ocean, the northwest Pacific Ocean, and the subtropical north Atlantic Ocean, is closely connected with the Tibetan Plateau–Lake Baikal dipole pattern of the atmospheric circulation and the NDVI over the eastern Tibetan

Plateau and Lake Baikal regions, implying that the synergistic SSTAs may contribute to the co-variability of the NDVI between the two regions through adjusting the atmospheric dipole pattern.

These findings imply that vegetation activity may not be only a local phenomenon in some areas, but also is likely to be remotely connected with the variations in vegetation over other regions, and atmospheric circulation anomalies are related to this co-variability. The co-variability of the NDVI between the eastern Tibetan Plateau and the Lake Baikal region suggests us to jointly protect and harness the ecological environment over the China-Mongolia-Russia economic corridor. Furthermore, it is necessary to analyze the teleconnection of vegetation across the globe and to reveal the features of the co-variability in global vegetation. These studies may further deepen the understanding of the co-variability of global vegetation and the associated reasons, providing a scientific basis for the organization and cooperation of protection and construction of the terrestrial ecosystem among countries and areas worldwide.

## Supporting information

**S1 Data. The data used for each figure and table in the Manuscript and Response to Reviewers and the data information.**
(ZIP)

**S1 File. Highlights.** Highlights in the article.
(DOCX)

## Acknowledgments

We thank the NASA's Global Inventor Modeling and Mapping Studies, the Global Precipitation Climatology Centre and the National Centers for Environmental Prediction for providing data. We are grateful to the anonymous reviewers for their constructive comments.

## Author Contributions

**Conceptualization:** Kejun He, Ge Liu.

**Data curation:** Kejun He.

**Funding acquisition:** Ge Liu.

**Investigation:** Jingxin Li.

**Methodology:** Ge Liu.

**Project administration:** Ge Liu.

**Supervision:** Ge Liu, Junfang Zhao.

**Validation:** Ge Liu, Junfang Zhao.

**Visualization:** Kejun He.

**Writing – original draft:** Kejun He.

**Writing – review & editing:** Ge Liu, Junfang Zhao.

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
