## [Decision Letter · Decision Letter 0]

26 Jun 2020

PONE-D-20-11620

Co-variability of the summer NVDIs on the eastern Tibetan Plateau and in the Lake Baikal region: associated climate factors and atmospheric circulation

PLOS ONE

Dear Dr. Liu,

Thank you for submitting your manuscript to PLOS ONE. After careful consideration, we feel that it has merit but does not fully meet PLOS ONE’s publication criteria as it currently stands. Therefore, we invite you to submit a revised version of the manuscript that addresses the points raised during the review process.

We look forward to receiving your revised manuscript.

Kind regards,

Broxton W. Bird

Academic Editor

PLOS ONE

Journal Requirements:

2. We note that Figures in your submission contain [map/satellite] images which may be copyrighted. All PLOS content is published under the Creative Commons Attribution License (CC BY 4.0), which means that the manuscript, images, and Supporting Information files will be freely available online, and any third party is permitted to access, download, copy, distribute, and use these materials in any way, even commercially, with proper attribution. For these reasons, we cannot publish previously copyrighted maps or satellite images created using proprietary data, such as Google software (Google Maps, Street View, and Earth). For more information, see our copyright guidelines: http://journals.plos.org/plosone/s/licenses-and-copyright.

1.    You may seek permission from the original copyright holder of Figures to publish the content specifically under the CC BY 4.0 license.

'This work was jointly sponsored by the National Key Research and Development

Program of China (Grant 2018YFC1505706), the Strategic Priority Research Program

of the Chinese Academy of Sciences (Grant XDA20100300), the National Science

Foundation of China (Grants 91637312 and 41775084), the Science and Technology

Development Fund of CAMS (Grant 2019KJ022), and the Basic Research Fund of

CAMS (Grant 2019Z008).'

'The author(s) received no specific funding for this work.'

Additional Editor Comments (if provided):

Reviewers' comments:

Reviewer's Responses to Questions

**Comments to the Author**

1. Is the manuscript technically sound, and do the data support the conclusions?

Reviewer #1: Partly

Reviewer #2: Yes

2. Has the statistical analysis been performed appropriately and rigorously? 

Reviewer #1: Yes

Reviewer #2: Yes

3. Have the authors made all data underlying the findings in their manuscript fully available?

Reviewer #1: Yes

Reviewer #2: Yes

4. Is the manuscript presented in an intelligible fashion and written in standard English?

Reviewer #1: Yes

Reviewer #2: Yes

5. Review Comments to the Author

Reviewer #1: The study attempted to analyze the covariability of NDVI between the eastern Tibetan Plateau (TP) and Lake Baikal in the Siberia. The authors analyzed the NDVI changes in the eastern TP and corresponding precipitation, and built a connection with remote area-Lake Baikal, in which presents an in-phase covariability on an interannual scale. Possible climate factors and atmospheric circulation associated with this were explored. The work is worth publishing before several revisions.

1. The reason for comparing the TP and Siberia is too weak, they should include these aspects, such as similar or different land scape, climate background, vegetation types, etc.

2. Line 90: should be “presented”.

3. How did you process the NDVI data except the Maximum value composite analysis? Since water body and bare ground exist in the two regions.

4. Data section, which data is used for solar radiation? Is that downward shortwave radiation or other else? Please add the data information.

5. The vegetation change is speculated to be modulated by atmospheric circulation anomalies, but how do you exclude the vegetation changes exert an influence on atmospheric circulation through the land-surface and atmosphere interactions, just like you mentioned sensible heat air pumping effect?

6. Line 116-117, a very related paper about precipitation in the TP and related mechanism (Precipitation over the Tibetan Plateau during recent decades: a review based on observations and simulations. International Journal of Climatology, 2018) would help you enrich your results and discussion.

7. Line 484: the equation for calculating SST index covers the NW-pacific, TE-pacific, and SN-Atlantic, but why is the northern India Ocean where exhibits significant negative correlations between summer dipole index and SST field excluded?

8. Line 508, the sentence is incomplete, please revise it.

9. Figure 1, the whole picture in terms of NDVI and precipitation should cover the study areas the TP and Lake Baikal simultaneously. The unit for precipitation may not be correct, please check it.

Reviewer #2: This paper depicts the co-variability of inter-annual vegetation variations between the eastern TP and LB; moreover, the co-variability is explained by large scale of atmospheric circulation viability. It is very interesting topic and it may be useful to understand the tele-connection of vegetation dynamics. But some minor problems should be addressed before acceptation.

1) In the Introduction and Discussion as well as Conclusion, the scientific implications and novelty of this study have not been clarified. In the revision, it is suggested to add more scientific implications on the climatology and ecology.

2) L79-82 presents that NDVI variations over the two regions were characterized by uptrend. However, Figure 3 shows that there are no a increasing trend. How to explain the paradoxes?

3) Please clarify the data source of solar radiation in the Table 1.

4) L484. Please clarify the physical implication of this SST index.

This study did not explore the possible physical links between SST anomaly and TP-LB dipole index. Is there possibly no physical link? Are they separated and derived from atmospheric oscillation and ocean oscillation, respectively?

6. PLOS authors have the option to publish the peer review history of their article (what does this mean?). If published, this will include your full peer review and any attached files.

Reviewer #1: No

Reviewer #2: No

---

## [Author Response · Author response to Decision Letter 0]

1 Aug 2020

Dear Editor Bird,

Thank you for your hard work and good advice.

We found your remarks and the reviewers’ comments to be extremely helpful for improving the manuscript. We truly appreciate your efforts and have revised our manuscript accordingly. Please refer to the specific responses to editors and to reviewers as follows. 

In addition, Dr. Jingxin Li participated in exploring the effect of the SSTAs in different oceans, so we listed her as the fourth author in this paper.

Best Regards,

Kejun He, Ge Liu, Junfang Zhao, Jingxin Li

Response to Editors:

Please ensure that your manuscript meets PLOS ONE's style requirements, including those for file naming. The PLOS ONE style templates can be found at https://journals.plos.org/plosone/s/file?id=wjVg/PLOSOne_formatting_sample_main_body.pdf and https://journals.plos.org/plosone/s/file?id=ba62/PLOSOne_formatting_sample_title_authors_affiliations.pdf

Reply: According to the requirements, we have revised the manuscript to ensure that the revised manuscript meets PLOS ONE's style requirements.

2. We note that Figures in your submission contain [map/satellite] images which may be copyrighted. All PLOS content is published under the Creative Commons Attribution License (CC BY 4.0), which means that the manuscript, images, and Supporting Information files will be freely available online, and any third party is permitted to access, download, copy, distribute, and use these materials in any way, even commercially, with proper attribution. For these reasons, we cannot publish previously copyrighted maps or satellite images created using proprietary data, such as Google software (Google Maps, Street View, and Earth). For more information, see our copyright guidelines: http://journals.plos.org/plosone/s/licenses-and-copyright.

 You may seek permission from the original copyright holder of Figures to publish the content specifically under the CC BY 4.0 license. We recommend that you contact the original copyright holder with the Content Permission Form (http://journals.plos.org/plosone/s/file?id=7c09/content-permission-form.pdf) and the following text: “I request permission for the open-access journal PLOS ONE to publish XXX under the Creative Commons Attribution License (CCAL) CC BY 4.0 (http://creativecommons.org/licenses/by/4.0/). Please be aware that this license allows unrestricted use and distribution, even commercially, by third parties. Please reply and provide explicit written permission to publish XXX under a CC BY license and complete the attached form.” Please upload the completed Content Permission Form or other proof of granted permissions as an "Other" file with your submission. In the figure caption of the copyrighted figure, please include the following text: “Reprinted from [ref] under a CC BY license, with permission from [name of publisher], original copyright [original copyright year].”

 If you are unable to obtain permission from the original copyright holder to publish these figures under the CC BY 4.0 license or if the copyright holder’s requirements are incompatible with the CC BY 4.0 license, please either i) remove the figure or ii) supply a replacement figure that complies with the CC BY 4.0 license. Please check copyright information on all replacement figures and update the figure caption with source information. If applicable, please specify in the figure caption text when a figure is similar but not identical to the original image and is therefore for illustrative purposes only. The following resources for replacing copyrighted map figures may be helpful: USGS National Map Viewer (public domain): http://viewer.nationalmap.gov/viewer/

Reply: Thank you for these detailed comments. We used OpenGrADS 2.0.a9.oga.1 to complete all the figures and downloaded it from the official website (http://www.opengrads.org) under the legal version.

3. Thank you for stating the following in the Acknowledgments Section of your manuscript: 'This work was jointly sponsored by the National Key Research and Development Program of China (Grant 2018YFC1505706), the Strategic Priority Research Program of the Chinese Academy of Sciences (Grant XDA20100300), the National Science Foundation of China (Grants 91637312 and 41775084), the Science and Technology Development Fund of CAMS (Grant 2019KJ022), and the Basic Research Fund of CAMS (Grant 2019Z008).' We note that you have provided funding information that is not currently declared in your Funding Statement. However, funding information should not appear in the Acknowledgments section or other areas of your manuscript. We will only publish funding information present in the Funding Statement section of the online submission form. Please remove any funding-related text from the manuscript and let us know how you would like to update your Funding Statement. Currently, your Funding Statement reads as follows: 'The author(s) received no specific funding for this work.'

Reply: Thank you for these detailed comments. The sentences in the Acknowledgments section have been revised. “We thank the NASA’s Global Inventor Modeling and Mapping Studies, the Global Precipitation Climatology Centre and the National Centers for Environmental Prediction for providing data. We are grateful to the anonymous reviewers for their constructive comments.” See lines 706–709 on p. 33 in the revised manuscript.

Response to Reviewer 1

We appreciate the detailed comments and suggestions from the reviewer, which were very valuable for improving the manuscript. In particular, the reviewer questioned why the SSTs in the Indian Ocean are excluded, which was very important and motivated us to further explore the potential mechanism of the SSTA forcing in different oceans. We have considered these comments seriously and revised the manuscript accordingly. The associated revisions in the manuscript are highlighted in red. Because the formulas and figures can not be displayed properly here, for complete responses to each question, please see the uploaded file named "Response to Reviewer 1".

Response to Reviewer 2

We appreciate the detailed comments and suggestions from the reviewer, which were very valuable for improving the manuscript. In particular, the reviewer expressed concern about the physical link between the Tibetan Plateau–Lake Baikal dipole pattern and the SSTAs in different oceans, which was very important and motivated us to further explore it in the revised manuscript. We have considered these comments seriously and revised the manuscript accordingly. The associated revisions in the manuscript are highlighted in red. Because the formulas and figures can not be displayed properly here, for complete responses to each comment, please see the uploaded file named "Response to Reviewer 2".

---

## [Decision Letter · Decision Letter 1]

19 Aug 2020

PONE-D-20-11620R1

Co-variability of the summer NVDIs on the eastern Tibetan Plateau and in the Lake Baikal region: associated climate factors and atmospheric circulation

PLOS ONE

Dear Dr. Liu,

Thank you for submitting your manuscript to PLOS ONE. After careful consideration, we feel that it has merit but does not fully meet PLOS ONE’s publication criteria as it currently stands. Therefore, we invite you to submit a revised version of the manuscript that addresses the points raised during the review process.

We look forward to receiving your revised manuscript.

Kind regards,

Broxton W. Bird

Academic Editor

PLOS ONE

Additional Editor Comments (if provided):

I agree with the reviewers' assessments of the manuscript. There are a few minor details to address, but the manuscript has been strengthened from its original form. In addition to the reviewers' comments, I would add that the NVDIs in the title be corrected to NDVIs.

Reviewers' comments:

Reviewer's Responses to Questions

**Comments to the Author**

1. If the authors have adequately addressed your comments raised in a previous round of review and you feel that this manuscript is now acceptable for publication, you may indicate that here to bypass the “Comments to the Author” section, enter your conflict of interest statement in the “Confidential to Editor” section, and submit your "Accept" recommendation.

Reviewer #1: All comments have been addressed

Reviewer #2: (No Response)

2. Is the manuscript technically sound, and do the data support the conclusions?

Reviewer #1: Yes

Reviewer #2: (No Response)

3. Has the statistical analysis been performed appropriately and rigorously? 

Reviewer #1: Yes

Reviewer #2: (No Response)

4. Have the authors made all data underlying the findings in their manuscript fully available?

Reviewer #1: Yes

Reviewer #2: (No Response)

5. Is the manuscript presented in an intelligible fashion and written in standard English?

Reviewer #1: Yes

Reviewer #2: (No Response)

6. Review Comments to the Author

Reviewer #1: The authors have well addressed the previous issues, however, there are still some issues needed to be answered before it is published. See below.

1. Line51, “…vulnerable to global climate change” would be better.

2. Line 262,263, statistical significance for the trend is needed.

3. Line 286, should be play important parts in.

4. Line 317, this can be removed.

5. Line 315-321, the author stated that precipitation corresponds more clouds and less solar radiation. However, there are calculated negative correlation between NDVI and precipitation, positive correlations between NDVI and OLR, and between NDVI and solar radiation. Why precipitation has different sign of correlation with that of OLR?

6. Line 341, why 200 hPa geopotential height instead of 500 hPa or other height is chosen?

7. Figure 6, time coefficients for EOF1 and EOF2 should be given accordingly.

8. Line 400, confused about the sentence: superimposition of the geopotential height anomalies regressed on the dipole index. Is the PC2 time coefficients regression on the eddy geopotential height field? Explanation is needed.

9. Except the climate factors, whether any anthropogenic activities have an influence on the changes in NDVI?

Reviewer #2: Authors have permfored important revisons, addressed my major concerns. I recommended the study for publication.

7. PLOS authors have the option to publish the peer review history of their article (what does this mean?). If published, this will include your full peer review and any attached files.

Reviewer #1: No

Reviewer #2: No

---

## [Author Response · Author response to Decision Letter 1]

25 Aug 2020

Response to Reviewer 1

We appreciate the detailed comments and suggestions from the reviewer, which were very valuable for improving the manuscript. We have considered these comments seriously and revised the manuscript accordingly. Our responses to the comments are summarized as follows. The associated revisions in the manuscript are highlighted in red.

1) Line51, “…vulnerable to global climate change” would be better.

Reply: This sentence has been revised. See line 51 on p. 3 in the revised manuscript. 

2) Line 262,263, statistical significance for the trend is needed.

Reply: Thanks for your suggestion. The NDVI over the Lake Baikal region shows a weak increasing trend of 0.11/decade, non-significant. The NDVI over the eastern Tibetan Plateau shows a decreasing trend of –0.33/decade, significant at the 90% confidence level. We have added the statistical significances in the revised manuscript. See lines 263 and 266 on p. 13.

3) Line 286, should be play important parts in.

Reply: This phrase has been corrected. See line 288 on p. 14.

4) Line 317, this can be removed.

Reply: According to this suggestion, the sentence has been removed. 

5) Line 315-321, the author stated that precipitation corresponds more clouds and less solar radiation. However, there are calculated negative correlation between NDVI and precipitation, positive correlations between NDVI and OLR, and between NDVI and solar radiation. Why precipitation has different sign of correlation with that of OLR?

Reply: The outgoing longwave radiation (OLR) is determined by temperatures at the top of cloud. Stronger (weaker) convection, associated with higher (lower) precipitation, can lead to more (less) cloud and drive the top of cloud to a higher (lower) level. When the top of cloud is higher (lower), lower (higher) temperatures appear at its top, leading to lower (higher) OLR. As such, higher (lower) precipitation/stronger (weaker) convection corresponds to lower (higher) OLR and solar radiation. Several previous studies (Kripalani et al., 1991; Chaudhari et al., 2018) have explained the precipitation-OLR anticorrelation. It is reasonable that precipitation has different sign of correlation with that of OLR. In the revised manuscript, we have briefly presented the OLR-convection/cloud relationship in Materials (see lines 142–143 on p. 7). In addition, we found a typo in the original manuscript, which may mislead the understanding of the reviewer. This mistake has been corrected in the revised manuscript (See line 323 on p. 16).

6) Line 341, why 200 hPa geopotential height instead of 500 hPa or other height is chosen?

Reply: We calculated the correlation between the Tibetan Plateau/Lake Baikal NDVI index and geopotential height field at different levels. The north-south dipole pattern related to the NDVIs is more significant and clearer at the 200 hPa level than that at the other levels. Also, the second EOF mode of geopotential heights displays a clearer north-south dipole pattern at the 200 hPa level. Therefore, we chose 200 hPa geopotential heights to reflect the typical atmospheric circulation pattern responsible for the co-variability of the NDVIs. Furthermore, Figure 7 indicates that the out-of-phase dipole pattern appears in both the upper and mid- to lower troposphere, forming a deep configuration of high-/low-pressure anomalies. This further suggests that the 200 hPa level is suitable for our study since it can also measure the atmospheric circulation anomalies at the lower level. 

7) Figure 6, time coefficients for EOF1 and EOF2 should be given accordingly.

Reply: Thanks for this suggestion. The time coefficients for EOF1 and EOF2 (i.e., PC1 and PC2) have been added in the revision as Figures 5c and 5d. The figure captions for Figures 5c and 5d have added in the revised manuscript accordingly (See lines 380–381 on p. 18 and lines 1055–1056 on p. 47). 

8) Line 400, confused about the sentence: superimposition of the geopotential height anomalies regressed on the dipole index. Is the PC2 time coefficients regression on the eddy geopotential height field? Explanation is needed.

Reply: Firstly, we calculated the 200 hPa geopotential height anomalies regressed upon the dipole index (i.e., PC2). Secondly, the geopotential height anomalies were superimposed on the climatological mean 200 hPa eddy geopotential height field. Through this superimposition, we can intuitively see anomalous characteristics of the atmospheric circulation systems (i.e., the South Asian high and the northeastern Asian blocking high) associated with this dipole. In the revised manuscript, we have presented the aforementioned explanation. See lines 402–406 on p. 19. 

9) Except the climate factors, whether any anthropogenic activities have an influence on the changes in NDVI?

Reply: We cannot exclude the influence of anthropogenic activities on the global and local climate and relevant changes in NDVI. However, we have reasons to believe that anthropogenic activities play an inessential role in modulating the variability of the NDVIs in the Tibetan Plateau and Lake Baikal regions. 

Firstly, the Tibetan Plateau and Lake Baikal regions are both sparsely populated areas. The vegetation in the Tibetan Plateau is relatively undisturbed by anthropogenic activities due to the low population (Zhang et al., 2013; Huang et al., 2016). Also, the population density in the Lake Baikal region is low (Zemskaya et al., 2020). This suggests that the changes in vegetation over the two regions may be dominantly affected by climate factors rather than anthropogenic activities. Secondly, the probability that the co-variability between the NDVI over the eastern Tibetan Plateau and that over the Lake Baikal region is triggered by simultaneous anthropogenic activities in the two regions is very low, especially on interannual timescales. 

Due to the abovementioned reasons, the influence of anthropogenic activities should not be considered as the main factor. So, we did not discuss the effect of anthropogenic activities in this paper. 

References

Chaudhari HS, Hazra A, Pokhrel S, Chakrabarty C, Saha SK, Pentakota S. SST and OLR relationship during Indian summer monsoon: a coupled climate modelling perspective. Meteorol. Atmos. Phys. 2018; 130: 211–225. https://doi.org/10.1007/s00703-017-0514-0.

Huang K, Zhang YJ, Zhu JT, Liu YJ, Zu JX, Zhang J. The Influences of Climate Change and Human Activities on Vegetation Dynamics in the Qinghai-Tibet Plateau. Remote Sens. 2016; 8(10): 876–893. https://doi.org/10.3390/rs8100876.

Kripalani RH, Singh SV, Arkin PA. Large scale features of rainfall and outgoing long wave radiation over India and adjoining regions. Contrib. Atmos. Phys. 1991; 64: 159–168. 

Zemskaya TI, Cabello-Yeves PJ, Pavlova ON, Rodriguez-Valera F. Microorganisms of Lake Baikal—the deepest and most ancient lake on Earth. Appl. Microbiol. Biotechnol. 2020; 104: 6079–6090. https://doi.org/10.1007/s00253-020-10660-6. 

Zhang L, Guo HD, Ji L, Lei LP, Wang CZ, Yan DM. et al. Vegetation greenness trend (2000 to 2009) and the climate controls in the Qinghai-Tibetan Plateau. J. Appl. Remote Sens. 2013; 7(1): 073572. https://doi.org/10.1117/1.JRS.7.073572.

---

## [Editor Report · Decision Letter 2]

8 Sep 2020

Co-variability of the summer NDVIs on the eastern Tibetan Plateau and in the Lake Baikal region: associated climate factors and atmospheric circulation

PONE-D-20-11620R2

Dear Dr. Liu,

We’re pleased to inform you that your manuscript has been judged scientifically suitable for publication and will be formally accepted for publication once it meets all outstanding technical requirements.

Kind regards,

Broxton W. Bird

Academic Editor

PLOS ONE
---

## [Editor Report · Acceptance letter]

16 Oct 2020

PONE-D-20-11620R2 

Co-variability of the summer NDVIs on the eastern Tibetan Plateau and in the Lake Baikal region: associated climate factors and atmospheric circulation 

Dear Dr. Liu:

I'm pleased to inform you that your manuscript has been deemed suitable for publication in PLOS ONE. Congratulations! Your manuscript is now with our production department. 

Kind regards, 

on behalf of

Dr. Broxton W. Bird 

Academic Editor

PLOS ONE